# Design Guidelines on LED Costumes for Dance Performances

**Ryo Izuta** [1,2,*]**, Tsutomu Terada** [1] **, Yutaka Yanagisawa** [2]**, Minoru Fujimoto** [2] **and Masahiko Tsukamoto** [1]

[1] Graduate School of Engineering, Kobe University, Rokkodai-cho, Nada, Kobe, Hyogo 657-0013, Japan; tsutomu@eedept.kobe-u.ac.jp (T.T.); tuka@kobe-u.ac.jp (M.T.)

[2] mplusplus Co., Ltd., Ebara, Shinagawa, Tokyo 142-0063, Japan; kani@mplpl.com (Y.Y.); m.fujimoto@mplpl.com (M.F.)

[*] Correspondence: r.izuta@mplpl.com

**Abstract:** We present design guidelines on light emitting diode (LED) costumes for dance performances assuming repetitive use during concerts. We used LED costumes more than 120 times for large concerts of well-known artists at venues of approximately 50,000 capacity that were commercially successesful and we updated the LED costume design twice based on our experiences during these concerts. Through analyzing the position of broken LEDs and the types of breakage and the problems that occurred during actual performances, we devised 17 design guidelines on LED costumes for dance performances. Thanks to these design guidelines, the LEDs on the costume are more difficult to break and we can prepare for any contingencies that may occur during a performance. We fabricated an improved LED costume based on our design guidelines and conducted endurance tests involving dancing. Throughout the endurance tests, the LEDs did not break, and other factors that cause LED breakage were found. We participated in two exhibitions to conduct special LED dance performances.

**Keywords:** design guideline; costume; LED; wearable computing; dance performance

## 1. Introduction

Live entertainment has been developing rapidly with advances in technology, and various technology-based performances have been produced. With rapid computer miniaturization and sophistication, costumes with wearable technology have been used in various performances. In particular, costumes on which illumination materials, such as light emitting diodes (LEDs), are sewn or pasted have been used in many live performances [1–3]. By controlling the timing/color of the lighting to the music and choreography [4], highly impactful performances can be staged. We previously fabricated an LED costume on which many LEDs are attached, and developed a live performance system for controlling the timing/color of the LEDs to the music and choreography. We also used LED costumes more than 120 times during large concerts of well-known artists at venues of approximately 50,000 capacity that were commercially successesful. Thus, a large amount of knowledge about using LED costumes were gained. We are using these LED costumes at live concerts that are being held now. If the detail of the LED costumes is described, it is possible to publish a part of the live concerts. Consequently, in testing out the LED costume design in actual concerts, we needed to conform to the non-disclosure conditions in ongoing concerts. Therefore, we describe the LED costume as possible. To achieve maximum performance when using an LED costume, the best LED positions and wiring for ease of dancing should be determined in terms of light expression without decreasing the quality as a stage costume. Also, in a dance performance with powerful movements, failure of LEDs and control devices cannot be avoided. Thus, it is necessary to have a costume design in which LEDs can be repaired promptly. Although

conventional LED costumes have been designed to prevent LED failures, no costume design takes LED maintenance into account. For making LED replacement easy, costumes with many long and narrow pockets for storing LEDs have been seen [1,5]. The front is fabricated of mesh, which allows seeing the light. Although conventional LED costumes must have sewn or glued LEDs, in this case LEDs are just put in these pockets. However, there are still many problems with conventional LED costumes such as having to take off the costume every time LEDs need to be repaired. Besides, popular artists hold concerts at multiple venues, and some artists hold concerts more often throughout a year. Therefore, designing a costume for long-term use without decreasing the quality as the stage costume is important. We updated our costume design from the experiments gained from using it in the above concerts. Hence, in this paper, we propose design guidelines for fabricating LED costumes for dance performances during large and long-term concert tours through our experiments. As follows, lighting costume technology is introduced in Section 2, and design guidelines for a wearable computer is described in Section 3. In Section 4, we describe the qualitative improvement of LED costume based on our experiences of using LED costumes at live concerts. In Sections 5 and 6, analyses of LED failures are conducted. Then, we propose the design guidelines on LED costume for dance performances in Section 7. In Section 8, the new LED costumes fabricated based on our proposal are introduced, and endurance testing involving dancing is conducted. In Section 10, dance performances using the LED costume fabricated in Section 8 are introduced. Finally, this paper is concluded in Section 11. Additionally, the contributions of this research are as follows.

- This paper presents guidelines for fabricating LED costumes based on the findings obtained by using our costume during large concerts.
- This paper clarifies the difference between using wearable technology in dance performances and daily life.
- The proposed design guidelines will enable designers and engineers to work together.
- Through using this LED costume design, we developed a new costume design that is both aesthetically pleasing and practical.

## 2. Lighting Costume Technology

Since 1184 when a dancer attached a battery to her waist to light a bulb on her tiara during the performance of *the ballet La Farandole* [6,7], stage performances that involve lighting up dancer's costume and accessories have been carried out. In this section, we introduce recent costume technology and stage performances with such costumes and discuss future costume technology.

### 2.1. Projection Mapping

Improvements in human-tracking technology have enabled the projection of an image onto costumes. Perfume [8], a Japanese pop group, conducted dance performance with projection mapping on their costumes in Cannes Lions 2013, which is one of the largest international festivals of creativity in the world. Projection mapping on a body must be tracked a body. The retroreflective material has the property of reflecting light in the direction of the light source, and when it is irradiated with infrared light, the reflection can be photographed with an infrared camera to obtain high brightness. Therefore, costumes, skin, and other spaces are identified from the difference in brightness of the reflection. To improve identification accuracy, the costumes were made of retroreflective cloth, which makes it easy to identify clothing, skin, and other space. Also, actuators were installed in their costumes, which open the costume so that the image will display when the members are not dancing and close the costume so as not to disturb the members when dancing. J. Lee et al. [9] implemented a projection system that irradiates images effects only on moving costumes and used it for a drama. Projection mapping can be used along with costumes performance and attracting attention as a new type of performance. However, since visual performances depend on projector ability, a high-intensity projector must be used. There are also problems such as it is not easy to track choreography with powerful movements,

resulting in a deviation between tracking and irradiating areas. Y. Watanabe et al. [10] implemented a projection system called *dynaFlash*. This system irradiates images while tracking a moving target at high speed, and can also irradiate images on the human body and clothing with a delay of only 3 ms at up to 1000 fps. However, the performance area is dependent on the projector and camera sensor position, which restricts the dancers.

## 2.2. Augmented/Virtual Reality

Costume performances with augmented reality (AR) and virtual reality (VR) have been investigated. Christoph et al. [11] proposed a mapping system that identifies human shapes and poses from an RGB camera image and maps a character's costume on the subject. They assume the use of commemorative photos, e.g., from theme parks, and do not assume the use of dance performances with powerful movements.

Also, performances which link an animation character on a screen to a real dancer by using motion data from smart wear, such as Xenoma [12], have been seen at many concerts. Since the smart wear is worn under the stage costume as innerwear, the quality of the dancer's costume does not deteriorate. However, AR/VR technology requires large screens, which inhibits the audience's view of the actual artists or dancer.

## 2.3. Wearable Devices

Since *cute circuit*, the first wearable technology fashion apparel brand in the world, was established in 2004, many fashions or costume designers have created costumes with wearable technology. By attaching electronic parts, such as E-Ink paper (electronic paper), electroluminescence (EL) wire, and LEDs, and by controlling the timing of lighting to the music, performances that were impossible before are now possible. Therefore, such costume performances have more high-generality than the costume performances picked up in this section and will spread in the future.

### 2.3.1. E-Ink Costumes

Dai Nippon Printing Co., Ltd. [13] unveiled an E-Ink dress that is made of E-Ink paper. By controlling the applied voltage, the color of the dress is changeable. It can also output patterns by controlling the attached E-Ink paper components individually. Since E-Ink paper does not emit light, i.e., we see reflected light, the change in colors can be seen even on a brightly illuminated stage. Moreover, it is not necessary to keep applying the voltage, which makes it an energy-saving operation.

However, E-Ink paper can display only two colors. Although the first look is interesting, the attraction after the first look deteriorates gradually. Moreover, if the E-Ink paper components are controlled individually, the granularity of expression increases in exchange for complex wiring, making maintenance difficult.

### 2.3.2. EL-Wire Costumes

EL wire is very flexible and easy to sew into costumes. By sewing EL-wire along the body line, it is possible to light up the silhouette of a dancer's figure, as if the dancer suddenly appeared from the dark. When multiple dancers wear the EL-wire costumes, by controlling the timing of light to the music and choreography a dancer moving instantaneously or as if the body was split can be seen. There are dance teams that mainly working with EL-wire costumes, e.g., EL SQUAD [14], a Japanese dance team, and Light Balance [15], an American dance team, are famous for intricate dance performances with EL-wire costumes.

EL-wire can also be used to draw a light curve, which is not possible with LEDs. Karpashevich et al. [16] created an interactive costume with EL wire that is based on Oskar Schlemmer's costume he made in the 1920s, and they generated new movements and choreography with it. Since one wire can light with only the same color, if multiple colors are used simultaneously, multiple light

sources must be controlled. Philknot [17], a Japanese costume manufacturer, solves this problem by weaving EL-wire and making layers.

EL-wire is inexpensive and has an AC drive, into which an inverter must be installed to control large devices. Organic EL-wire does not require an inverter, however, it is just expensive. Light power decreases the further from the light source with this costume. Therefore, there are still issues with EL-wire costumes regarding dance performance.

### 2.3.3. LED Costumes

Rosella and Genz [18], fashion designers at *CuteCircuit*, presented new costumes with LEDs in 2004. Since then, many types of LED costumes have been fabricated. In 2008, Chalayan, a British fashion designer, presented the *sparkling crystal dress* [19] with LEDs and laser diodes. LEDs are installed inside Swarovski crystals, which do not lose their beauty. These costumes are extremely delicate; thus, exaggerated movements, such as dance, cannot be done because wearers need to stop when the LEDs and laser diodes are on. Such as *CuteCircuit* costume [18] and *sparkling crystal dress* [19] fabricated by Chalayan, costumes onto which the LEDs are sewn have been fabricated over ten years ago. However, there were many challenges in using such a costume for dance performance due to the above movement limitations and the fact that the wearer must carry a large battery.

With advances in wearable technology, LED costumes can be used at venues such as theme parks and music concerts. In the electrical parades held at Disney theme parks [3], large LEDs are arranged on the dancer's costume to avoid more movable body parts. Fujimoto et al. [4] developed an LED-costume-control system for dance performances. At live events, LEDs on the costumes turn on in a pre-programmed pattern, which is very simple, but thanks to their performance system, more delicate performances are now possible.

LED costumes have rapidly evolved due to the appearance of LED strips. LED strips enable the easy installation of many LEDs on a costume, providing a more impactful performance. In the Super Bowl half time show in 2011, the National Football League's large event, Black Eyed Peas [2], an American pop group, wore LED costumes and performed with more than 100 dancers wearing LED bodysuits. A control device was placed on the dancer's waist so the dancers could turn on/off the LEDs. Some of the dancer's LEDs lit with different colors due to LED breakage. Thus, all LEDs of one dancer are controlled by one device. LEDs were arranged on the more movable parts of a dancer's body such as waist and knees, which was problematic. If there are LEDs on the joints such as elbows and knees, the LEDs are bent and stretched frequently, so there is a high possibility that the LEDs break. The main artists also wore special LED costumes that resemble American football protectors and LEDs were fixed to avoid LED failure. Katy Perry [1], an American pop star, wore a beautiful LED bodysuit on the *American Idol*, an American music TV program. This LED bodysuit was created by *CuteCircuit*. LEDs were arranged on the suit to avoid the more movable body parts. Retroreflection cloth was used except for the LED areas, which shone when illuminated by stage lights. Since the LED wiring is hidden in such a costume and there are not slits for repairing LEDs, a wearer must take off the suit when LEDs need to be repaired. In the keynote session of TAIT [20] held at CES 2018 [21], performers wearing LED bodysuits conducted a trampoline performance. The LED units were arranged on the suit at regular intervals, which made it easy to replace broken LED units. Seen from the performance video [21], only one chest LED unit broke, though another LED was functioning normally. Therefore, the LEDs were controlled by multiple threads (the end of the LED unit might also have been broken).

Since LED strips make it easy to arrange many LEDs on a costume, video images can be output. *Cute circuit* [22] presented an LED dress made with many LED strips in 2010. Although the sense of drape that is the unique point of the dress has been lost, impactful performances are possible because it can output video images at high resolution. In 2017, Komaden [23], a Japanese lighting material company, presented an LED costume, called *LED Video Costume*, on which LED display units were arranged. The shape of the display is a triangle, enabling the arrangement of LEDs in three dimensions such as the surface of a costume. However, since the LED display is larger than LED strips

or LED bulbs, it is limited to where LEDs can be placed. If LED display units break, it is noticeable. Berglund et al. [24] fabricated an interactive LED dress that can change its color with a fairy-tale-like interactive wand. They discussed the effective light diffusion and LED layout to make the whole dress glow evenly. They also proposed two methods of arranging LEDs for durability. Hardy et al. [25] developed a yarn in which LEDs are woven, called *LED-Yarns*, and used it for a carnival costume. LED-Yarns is aesthetically pleasing on its own, so it can be embroidered on costumes. It has some elasticity, which provides durability for intricate movements such as in carnival dance.

*2.4. Summary*

Costume lighting technology has various purposes: as just a costume, a display, however, we considered such technology for use at large-scale concerts involving many dancers. An example of dance performance using lighting costume technology is shown in Figure 1. We conducted a 5-min dance performance at the international conference of SIGGRAPH ASIA held in Japan [5]. In this dance performance, ten dancers wore the same LED costume and controlling the color of LEDs and timing of lighting to the music, and the dance performance was augmented. For example, lighting LEDs to a piece of choreography, such as the ten dancers linking with each other, it is possible to conduct a dance performance that cannot be done by only one dancer. For example, only arms other than the main dancer's arms are lit, and they are arranged in a line, thus a dancer can dance as if the dancer have many arms. In this performance, 10 LED costumes were controlled, though we have controlled 30 to 130 LED costumes at other concerts.

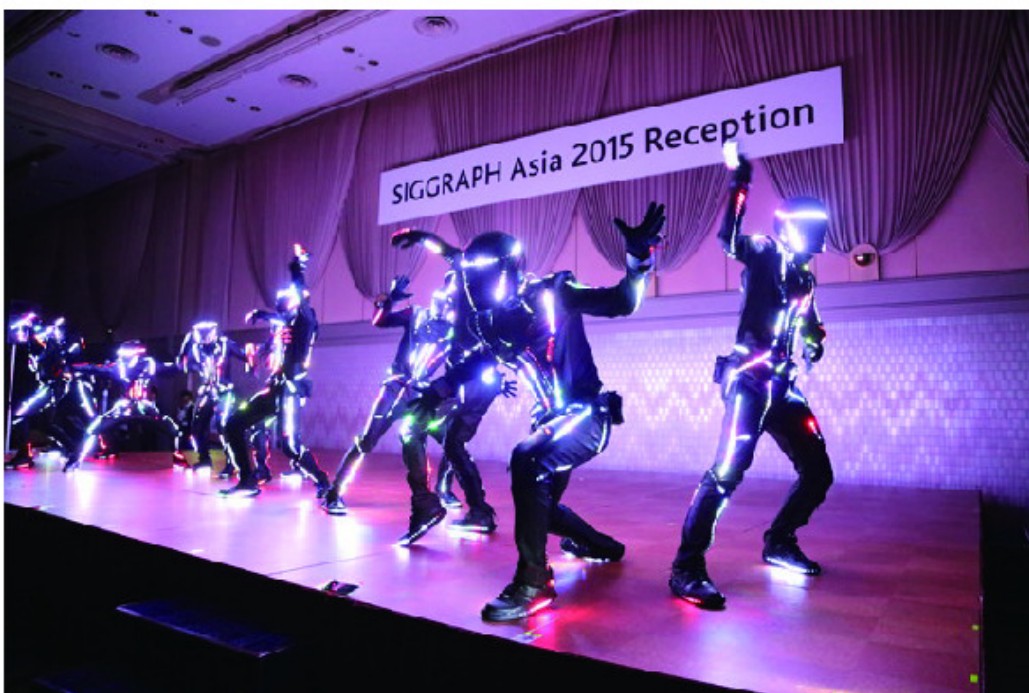

**Figure 1.** Light emitting diode (LED) dance performance for SIGGRAPH ASIA 2015.

In this study, we did not use lighting costume technology just for lighting costumes but also for enhancing the dance performance, as shown in Figure 1. As mentioned in this section, there are many ways to make costumes and light up dancers. Before, we could do nothing more than light up a costume with luminous paint, however, thanks to advances in information technology, electronic devices such as EL-wire and E-Ink paper can be attached to a costume and their light or color are controlled. However, these devices can be only turned on/off or change color. With an LED costume, color and lighting timing of LEDs can be controlled separately. Hence, multiple meanings can be added to one piece of choreography or movement by lighting. Since the light intensity of LEDs is stronger

than with other electronic devices, dances look more dynamic. However, when using LEDs, since the light from each LED is seen directly, the light from each LED diffuses and overlaps. For this reason, the LEDs placed on the fingertips overlap with each other, and the movement of each finger cannot be understood. When using EL wires, we are looking indirectly at the light passing through the wire, therefore the light is difficult to diffuse and can be seen as a wire shape. Therefore, you can see the movement from the LED even if you put it on your fingertip. Also, EL-wire is lighter and softer than LEDs, and it can do delicate actions, such as drawing curves, making human-shaped silhouettes with wiring, and being wired up to the fingertips. LEDs can be wired only linearly. Therefore, it is difficult to reflect such delicate movements with an LED costume. If more LEDs are arranged on a costume, more delicate movements can be expressed. However, the costume becomes heavy, and the risk of LED failure increases.

On the other hand, projection mapping and AR/VR with equipment on the stage can enable delicate expressions such as tracking the fingertips of a dancer. Considering the accuracy of tracking and the irradiation range of the projector, only a small number of dancers can perform. To track human skeletons, camera devices are needed. However, even kinect [26] can recognize up to six people at a time. Therefore, for example, tracking 100 people and projecting them requires more than 17 kinects, and a larger tracking system must be built, which is not practical. Despite these limitations, LED costumes, projection mapping, and AR/VR will be considered useful in many concerts in the future. LED costumes are good for dynamic performances with many people, though some subtlety is applied. On the other hand, projection mapping and AR/VR are suitable for delicate performances with a small number of people, but it is difficult for a large number of people to make a dynamic performance with a large concert venue. The concert targeted in this study is a large-scale concert at a venue of approximately 50,000 capacity and involved 10 to 100 dancers. Although the costumes become heavy and it is difficult to produce delicate effects such as projection mapping, we focus on the development of LED costumes, which can add various meanings to choreography with the lighting and can be used for many dancers.

As mentioned in this section, LED costumes are used in many events, however, most are one-off events. They are sometimes used for long-term concert tours, but specialized staff with knowledge about the costumes, engineering, and concert operation, etc., are needed, and they have to cope with many failures, which is expensive. When using LED costumes in dance performances, LED and control-device failures cannot be avoided; hence, costumes are designed that can be quickly repairable even just before the act. Such concert costumes cannot be fabricated until the concept or direction of the concert is decided, thus the fabrication time is very short, and we do not have any time to think about creating costumes taking into account repair and operation. Therefore, when creating LED costumes specialized for dance performances, it is important to be aware of issues from past concerts and incorporate the solutions into the costume design at the time of the update with a short delivery time and without decreasing the quality as stage costumes, which is the best way to make long-term use efficient. We used LED costumes more than 120 times in large-scale concerts at venues of approximately 50,000 capacity during the last 5 years and updated the LED costumes twice based on these experiences. It is thought that these experiences are useful for devising our design guidelines.

## 3. Conventional Design Guidelines for Wearability

Before proposing our design guidelines for LED costumes, we discuss previous studies on LED costumes. LED costumes are fabricated using wearable computing, and the design guidelines for wearable computing, which are related to our design guidelines, are often updated. Birringer et al. [27] conducted an experimental performance with technology such as wearable devices and projection mapping for the project called *Design in Motion*. They introduced the design process of the performance, however, they did not discuss the detailed design process of the wearable devices. Also, they also did not discuss how to maintenance. There are many points to consider when using wearable computing such as wearability, wearer movements, and sensor position, and design guidelines about them have

been proposed. In 1998, Gemperle et al. [28] proposed design guidelines for using wearable devices, specifically regarding the preferable position and shape of the device considering the wearability and wearer movements. They discuss 13 guidelines to consider for using wearable devices such as placement, form language, and human movements. Since the manner of using a wearable device has changed with developments in wearable computing, these design guidelines have been updated. In 2014, Motti et al. [29] proposed design guidelines for wearable devices and argued that such devices should be designed with an emphasis on comfort, affordability, aesthetics, and wearability. Zeagler [30] proposed new design guidelines based on wearable design for the past 20 years such as that by Gemperle et al.'s. He listed 13 guidelines such as proxemics, weight, and distribution. The detailed content of each point is published on his web site [31]. Many design guidelines for wearable computing focusing on clothes were also proposed. Koon et al. [32] created a driven wing for theatrical performances and summarized the design constraints. They created the wing based on four points, silent actuation, light and compact, reliable and user-friendly design, and dramatic wingspan. Walter Lee et al. [33] developed an inner shirt with an EMG sensor to analysis injuries that occur while wearing a spacesuit and they focused on six design guidelines such as reliable and mechanical coupling. Lee et al. [34] fabricated a jacket for cyclists with LEDs installed and introduced design guidelines focusing on getting on a bicycle. Greinke et al. [35] designed a shirt called *Solar Shirt*, which is sustainable and ecological. The goal is establishing zero energy wearable computing, and the shirt has a solar panel power-generation mechanism in the design.

The design guidelines for wearable computing are described in great detail; therefore, they should be selected according to the environment, purpose, gender, and so on. The list of general design guidelines points often mentioned in previous studies are shown as follows.

- Placement: By attaching a sensor to the body as described in literature [30], it is possible to acquire body movement data, however the position of the sensor must be determined in consideration of the body movement. For sensing, the sensors should be fixed at a proper position.
- Sizing: The wearable device should be as small as possible. If the presence of the wearable device makes the wearer uncomfortable, it should be redesigned, for example, separating the battery from the device.
- Human movement: The wiring and wearable device position should be determined after considering the wearer's movements. Wiring that inhibits movement results in problems such as disconnection.
- Proxemics: Wearable devices should not be perceived.
- Aesthetics: Wearable devices are used in various environments; hence it should be designed according to the use environment.
- Long-term use: It is necessary for a design to not stress the wearer even when worn for a long time.
- Washable: Electrical devices should be easily removable for washing the clothing.
- Reachability: The wearer often the device turns on/off and adjust the sensor position; therefore, it should be within reach.

LED costumes for repetitive use in dance performances should be fabricated along with the above conventional design guidelines, but there are points that do not need to be considered such as sensor position since LED costumes do not have sensors. Previous design guidelines often assume use for daily life; hence, they cannot be applied in unusual environments such as live entertainment. In the entertainment environment, the production effect is given the highest priority, therefore the proxemics and human movement that are prioritized in other environments are often sacrificed, which requires effort from dancers. Therefore, design guidelines for the entertainment environment are needed, and fabricating and using LED costumes in large-scale concerts for five years greatly contributes to the creation of the design guidelines for a dance performance.

## 4. Qualitative Improvement of Costume

Experience of the use of LED costumes at live concerts is important for the fabrication of LED costumes. In this section, we describe the qualitative improvement of LED costumes based on the experiences of their use at live concerts.

We are using these LED costumes at live concerts that are being held now. If the detail of the LED costume is described, it is possible to publish a part of the live concerts. Consequently, in testing out the LED costume design in actual concerts, we needed to conform to the non-disclosure conditions in ongoing concerts. Therefore, we describe the LED costume as possible.

We have used our LED costumes more than 120 times at large-scale concerts in venues with approximately 50,000 capacity for 5 years, and more than 100 times per year at small-scale concerts, music TV programs, and music videos. We used 10 to 130 costumes depending on the scale. We did not only fabricate the LED costumes but participated in the concerts as concert staff. We dealt with various problems, such as LED failures and control-device failures, which occurred during the shows. In large-scale concerts, we have seen how the LED costumes are handled other than during dance such as changing quickly, transporting costumes, and suddenly changing dancers. Hence, we obtained knowledge of not only the use of LED costumes in dance performances, but also of their handling from costume transportation to maintenance. In particular, we obtained a large amount of knowledge regarding LED failure, since they occurred at every concert, and updated the costumes based on these experiences. The LED costume used for a certain concert was designed by a costume designer without any engineering knowledge. However, many companies are involved in the fabrication and production of LED costumes. Therefore, there is a confidentiality obligation regarding the technical part. In this paper, we discuss the fabrication of LED costume and the operation during concerts, except for the confidential content. A designer often sacrifices safety and maintainability to make an interesting design. An LED costume that is easily damaged leads to an increase in maintenance costs. Therefore, arranging the LEDs so that they would be difficult to break while maintaining the quality of the costume is important. We updated our LED costume twice between 2013 and 2015 and used three versions of the LED costume for the concerts for three years. Three sections of our costume are shown in Figure 2. All costumes have long and narrow pockets that are made by mesh fabric to store LED units. The arm, thigh, and leg have wide moving ranges, and the LED around them is bent and stretched repeatedly, which causes LED breakage. Thus, the LED layout was redesigned. The costume created in 2013 is our first costume, and we faithfully accepted the designer's intention because we did not have data on what type of LED layout results in breakages. Ten of these costumes were fabricated, and used for approximately ten-minute dance performances during concerts that lasted approximately 3 h, held 25 times for 6 months with 50,000 concert goers. LED strips were used for this costume, which was unitized for easy repair. We used addressable LED strip [36]. The LED source is a 5050SMD built-in SK6812 chip, PCB thickness is 0.25 mm, and 60 LEDs are installed per one meter. The 3-pin plug housing JST SM connector [37] is soldered to input, and the 3-pin receptacle housing JST SM connector is soldered to the output. The surface of the LED strip is coated with silicone resin for waterproofing, and each unit is connected by SM connector so that it can be partially replaced, and all cables are soldered manually. This costume has long LED units that wind around the body, which caused breakages. Four pockets were sewn inside the costume, and devices and batteries were stored in these pockets. These were located on the chest and hip so as to not disturb the dance. We fabricated 10 of the costume created in 2014 for two dance performances, one was about 6 min and the other was about 4 min and 30 s, during concerts that lasted 3 h. The concert was held 30 times for 4 months with approximately 15,000 concert goers. The LED units for this costume were designed to be short based on our experiences from the first costume. In addition to LED strip 5050SMD built-in SK6812, ring-shaped addressable LED strips were used. Its LED source is the same as the LED strip we use in 2013, and it consists of 16 LEDs and is arranged in a ring shape. The PCB thickness is 1 mm. The inner diameter is 30 mm and outer diameter is 45 mm. It can connect to the other LED units with JST SM connector. Due to the wiring placing and ring-shaped LED units at the body joints, such as elbow,

thigh and hip, and knee, LED breakages were decreased. The longest LED unit, the number of LED is 28 and the length is approximately 47 cm, was placed from the knee to ankle as the designer intended; breakages frequently occurred since the knee was repeatedly bent and stretched while dancing. The average length of the LED unit used in the costume is approximately 16 cm and the second longest LED unit is approximately 38 cm. Therefore, LEDs that are too long are easy to break. Since the devices and batteries located on the chest in the 2013 costume disturbed dance, all devices and batteries were installed behind the waist. The costume created in 2015 was a jumpsuit as the designer intended and 30 pieces were fabricated. Twenty pieces were for men and the others for women. All costumes had the same LED layout, but the end of LED units in the women's costume, such as at the arms and legs, were shorter than in the men's costumes. These costumes were used for approximately 27-min dance performances during concerts and approximately 20-min dance performances during other concerts. Both concerts were approximately 3 h long with approximately 50,000 concert goers. One was held 21 times and another was held 20 times from 2015 to 2016. Based on past breakage experiences, the LED layouts on the arms and leg were simplified, and the LED units were separated at the joints of the body, decreasing breakages. Since the device and battery positions fabricated in 2014 were gathered in one place, the waistline silhouette collapsed. Therefore, they were distributed over the shoulders and hip, which would not interfere with the dance.

We updated these costumes by improving the LED layout and wiring and used it during large-scale concerts from 2013 to 2015. Although the costume design itself is important, LED layout constraint to reduce breakage is also important from the point of long-term use and production cost. These are qualitative improvements, and what type of dance leads to LED breakages was not determined. Therefore, more quantitative analysis is required to fabricate sturdier LED costumes. In addition, we take into consideration every type of handling from transportation to maintenance, which is extremely rare. The knowledge obtained from these experiences will be useful in creating our design guidelines.

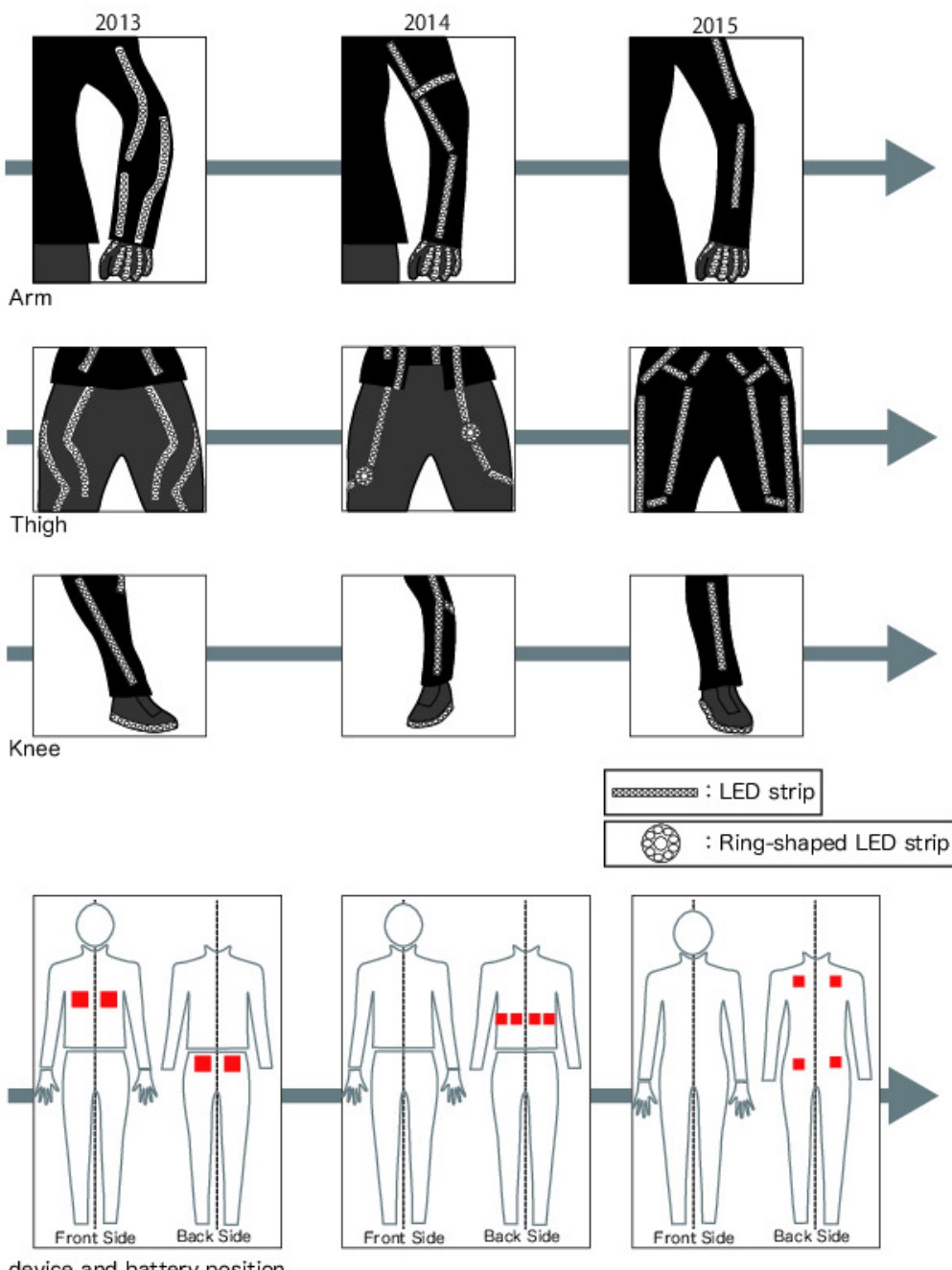

**Figure 2.** LED layouts on 2013 to 2015 costumes.

## 5. Analysis of LED Breakage

In this section, we discuss the LED wiring on the costume, which is an important factor in proposing design guidelines on LED costumes for dance performances. We updated our LED costumes twice and fabricated pieces and notified the costume designer of the constraints of LED wiring based on our experiences from past concerts without quantitative analysis. Although the number of LED failures has decreased, many staff members are needed for costume maintenance. To fabricate a costume that is sturdy against breakage, more detailed analysis is necessary. Therefore, we collected LEDs that broke during concerts and analyzed them.

## 5.1. Procedure

We used the LED costume fabricated in 2015 shown in Figure 2 during a large concert tour, and analyzed how LEDs break and how the wiring is prone to breakage. We used 20 of the 2015 LED costumes during the concert held seven times from April to July 2019, and collected LEDs that broke during the concerts. Concretely, when an LED is broken, the position of the broken LED is recorded, and it is collected and analyzed later. There were around 50,000 concert goers for each concert, and the concerts lasted approximately 2 h and 30 min. The performance using LED costumes was approximately 23 min. Data on the type of LED breakage and locations of broken LED units on the costumes were collected.

## 5.2. Result

The types of LED breakage are shown in Table 1. The number of broken LED units was 149, which is 21 per concert on average. We have also developed LED strips along with the improvement of LED costumes since 2013; hence, our LED strip filled with silicone resin on the surface is used, as shown in Figure 3. In Table 1, there are 6 broken types not including unknown. Fracturing, peeling off of LEDs, and rending of cable occurred while dancing. Poor soldering and poor connector were defective LED units. The most common type of breakage was fracturing. This is considered due to the LED unit being bent repeatedly during dancing. The second most common type was peeling off of LED. This was also caused by dancing; however, the heat and sweat of the dancer also lead to this type of breakage. These LEDs had water drops between the parts. Once an LED is submerged, it is easy to peel off from the circuit board and deteriorates quickly. Tear-off the cable also occurred. Each LED unit is connected with a cable, but if the cable is entangled or does not fit in the specified location, it will be torn off during the dance. Crushing also occurred but not due to dancing, but by the dancer hitting something on the stage.

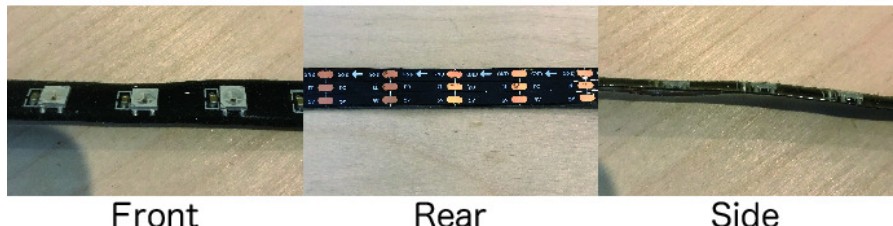

**Figure 3.** Appearance of LED strip used for LED costume.

**Table 1.** Types of LED breakage.

| Breakage Type | Number |
| --- | --- |
| Fracturing | 86 |
| Peeling off of LED | 27 |
| Torn-off cable | 17 |
| Crushing | 6 |
| Poor soldering | 4 |
| Poor connector | 3 |
| Unknown | 6 |

The LED layout on the costume and the positions of the broken LED units are shown in Figure 4. The left side is the LED layout, which included 907 LEDs and 62 LED units. Each LED unit is denoted with an arrow. The right side is the positions of broken LED units, represented with $X^N$ symbols. The notation $X^N$ symbol denotes the broken LED position, and $N$ denotes the number of the broken LED unit. The LED layout on this costume is symmetrical. Depending on the dance, there is a possibility

that the degree of breakage on both sides is different. However, not the influence of the dance, but the influence of the design of the costume itself, is being investigated. Therefore, we marked the broken LED position only on the right half of the body. The most broken position was around the hip, on which there were 36 instances of broken LED units. This was because they were bent repeatedly during dancing. The LED units around the front thigh also fractured frequently. There were 17 LEDs in these LED units, which was longer than the other LED units and frequently bent and stretched while dancing, which caused fractures. The LED units around the tips of the arms and legs frequently broke. This was due to the body size of the dancer. When a shorter dancer wore a large LED costume, those LEDs fractured. In particular, the LED unit on the leg was so long that it bent around the ankle. During long concerts, the dancers who were going to wear LED costumes often changed by the concert date. Therefore, shorter dancers may sometimes have to wear a little larger LED costumes.

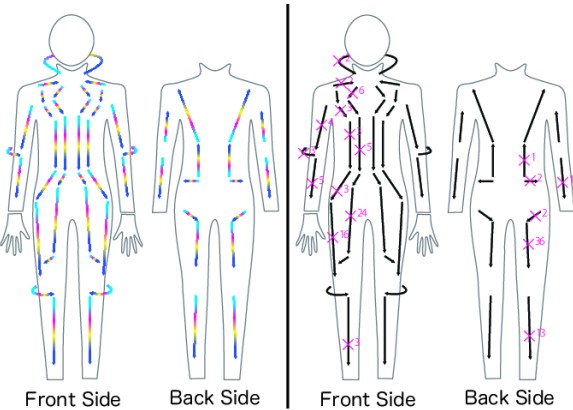

**Figure 4.** Left: reference of LED layout. Right: positions of broken LED units.

## 6. Endurance Test of LED at Joints

Dance involves very complex movements and it is difficult to observe how LEDs become damaged. Therefore, we conducted an endurance test for bending and stretching of LEDs with human movements. Although the actual endurance experiment is performed mechanically, this time, because it is assumed to be used for dance performance, an experiment reflects the movement of the body. The LED units were attached to the body parts that move significantly and are bent and stretched repeatedly. From our experiences on using LED costumes during past concerts, we found that the LED units attached to the elbow, knee, thigh, and hip easily break; hence, an endurance test on these three body parts was conducted.

### 6.1. Procedure

The LED units were attached to the elbow pad, knee pad, and short pants, as shown in Figure 5. A participant wears them, and bends and stretches until the LED units break. One trial of the elbow is shown as follows.

Step 1. A participant starts when the elbow is stretched.
Step 2. A participant bends it as much as possible.
Step 3. A participant restoring it to its original state. (Return to step 1.)

One trial of the knee, thigh and hip is shown as follows.

Step 1. A participant starts from a standing state.
Step 2. A participant sits on the chair with his/her knees bent almost right-angle.
Step 3. A participant stands up again. (Return to step 1.)

These exercises were repeated until the LED units broke. The number of participants is one, and the participant takes a 5-min break every 200 times of bending and stretching.

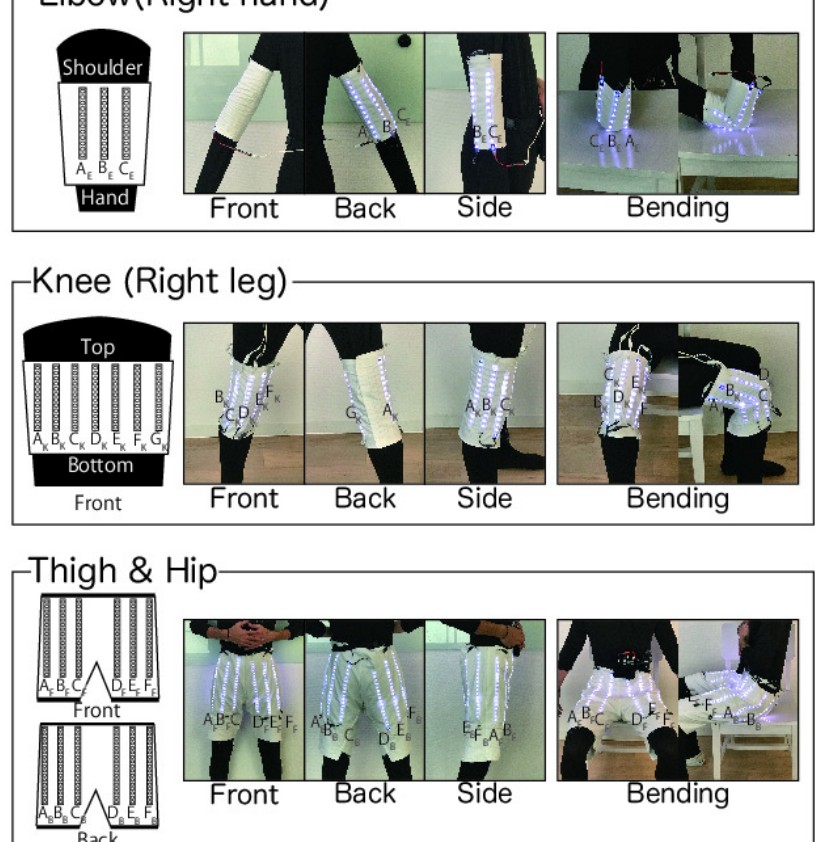

**Figure 5.** LED clothing for endurance test. Left: The appearances of the clothing and LED positions. Right: Appearances when wearing and bending.

*6.2. Result*

The number of trials when each LED unit fractured is shown in Tables 2–4. The symbols of position in the tables correspond to the symbols that represent the position of an LED unit in Figures 6–8, which are shown in Figure 5.

**Table 2.** Number of trials until fracture (elbow).

| Position | Trials |
|:---:|:---:|
| $A_E$ | 1229 |
| $B_E$ | No breakage |
| $C_E$ | 1861 |

**Table 3.** Number of trials until fracture (knee).

| Position | Trials |
|----------|--------|
| $A_K$ | 2033 |
| $B_K$ | 2073 |
| $C_K$ | No breakage |
| $D_K$ | No breakage |
| $E_K$ | No breakage |
| $F_K$ | 2009 |
| $G_K$ | 2242 |

**Table 4.** Number of trials until fracture (thigh and hip).

| Position | Trials | Position | Trials |
|----------|--------|----------|--------|
| $A_F$ | 639 | $A_B$ | 1600 |
| $B_F$ | 237 | $B_B$ | No breakage |
| $C_F$ | 991 | $C_B$ | No breakage |
| $D_F$ | 1222 | $D_B$ | No breakage |
| $E_F$ | 333 | $E_B$ | No breakage |
| $F_F$ | 399 | $F_B$ | No breakage |

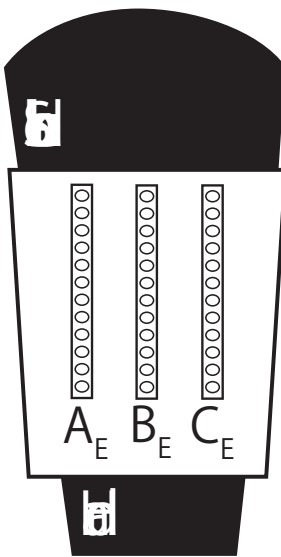

**Figure 6.** LED positions of elbow.

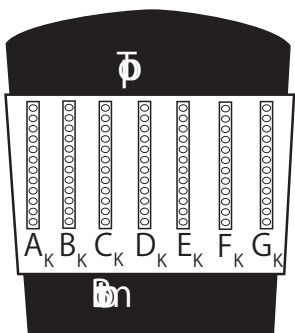

**Figure 7.** LED positions of knee.

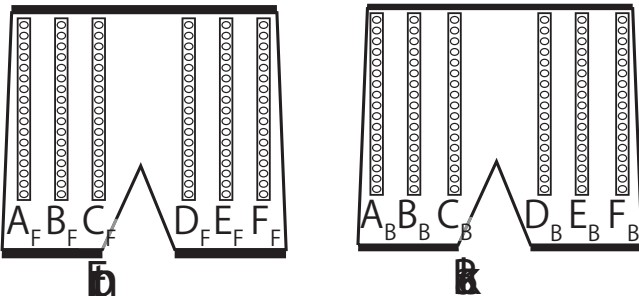

**Figure 8.** LED positions of thigh and hip.

The damaged LED was clearly identified at the beginning of the experiment. Therefore, this is the number of times that the damaged LED was broken. The elbow exercise was 2000 times, knee exercise 2500 times, and thigh and hip exercise 2000 times. The LED units that did not break are described as 'No breakage'. The LED units located on the side of the elbow, $A_E$ and $C_E$, and on the side of the knee, $A_K$, $B_K$, $F_K$, and $G_K$, fractured at the elbow and knee joints, and pressure was applied to the inflexible short side direction of the LED unit, which caused the fractures. The LED units located at $B_E$ of the elbow and $C_K$ and $D_K$ of the knee did not break. These LED units were attached along the direction of the bending joints. At the thigh and hip, all LED units located at the front of the body broke. The front of the short pants wrinkled at the groin, and pressure was applied to the LED units because of that wrinkle. When standing up, the upper body has to be bent a little forward, which also puts pressure on the LEDs located at the groin. At the hip, the LED units except $A_B$ did not break. These LED units were located along the direction to the bend in the hip and were not damaged. For the same reason as the elbow and knee, the LED unit located at the side of the joint easily fractured. This is why the LED unit located at $A_B$ fractured. Although the LED unit at $F_B$ symmetrical with that at $A_B$ did not break, it was more damaged than the others due to repeated bending and stretching.

In this test, it was found that the LED unit located at the side of the joints ($A_E$, $C_E$, $A_K$, $B_K$, $F_K$, and $G_K$) and around the groi(from $A_F$ to $F_F$) easily fractured. Therefore, when creating an LED costume, it is necessary to avoid arranging the LEDs in the fracture positions founded during this test or to arrange them to avoid applying excess pressure. We also found that a crease was made once the LED unit was significantly bent, and bending was repeated at the same crease. Repeatedly bending in the same crease results in fractures. This is considered to be due to the softness of the LED strip. Therefore, it is necessary to devise a stronger LED unit.

## 7. Design Guidelines for LED Costumes

Based on the analysis of broken LEDs, the above endurance test, and our experience of using LED costumes for the large-scale concerts since 2013, we propose 16 design guidelines on an LED costume for dance performances assuming repetitive use during large-scale concerts.

1. Safety: The LED strip circuit board and connector are sharp, and they may wound the dancer depending on the LED layout on the costume. Therefore, the costume must be designed taking into account dancer safety.
2. Easy to wear/take-off: During a concert, dancers sometimes have to change costumes quickly. Therefore, LED costumes must be easy to put on and take off.
3. Easy-to-change LEDs, control-devices, and batteries: LED units, control-devices, and batteries have to be easily changeable or repairable even after the dancer has worn the LED costume.
4. Easy to dance: The costume should not interfere with the dance performance due to the attached LED units and control-devices on the costume.
5. Control-device position: Dancers perform dramatic moves, and such movements cause failures. The control-device should be mounted where it does not move much.

6.  Control-device size: The control-device should be as small as possible for dance performances. The battery is the major component of such devices; thus, it is also important to consider the battery size with the lighting time of LEDs.

7.  Number of control-devices: If all LEDs on a costume are controlled by one device, all LEDs go out when the device breaks. Hence, LEDs should be controlled by multiple devices.

8.  LED layout constraints: An LED layout in which LEDs easily break was found from the analysis discussed in the previous section. An LED location prone to failure should be avoided.

9.  LED-unit length: A long LED units easy fractures; thus, a long LED unit should not be used on a costume.

10. Drip-proof or waterproof: Since dancers sweat a lot while dancing, LED units attached to a costume must be drip-proof or waterproof.

11. Adjustable: When a shorter dancer wears a larger costume, the LED units on the arm and leg often break. It is necessary that the LED-unit position can change according to the shape of the dancer, or the costume has an adjustment function.

12. Fitting: The number of broken LED decreases when a dancer wears an appropriately sized costume.

13. Moderately flexible LED strip: LED strips for LED costumes should be moderately flexible so that they get no crease.

14. Aesthetics: LED costumes should be used as a normal stage costume when the LEDs turned off.

15. Washable: A dancer sweats a lot during dancing. Therefore, the LED costume should be washed after finishing the concert. Alternatively, all LEDs on the costume should be easy to remove.

16. Transportation: Careful transportation of the LED costume is necessary. For example, if it is folded and packed in a costume case, LEDs may bend due to the weight of the costume. To avoid failures during transportation, LED costumes should be transported while hanging according to the size of the costume.

The design guidelines proposed in this paper are the items derived from the analysis and endurance experiments performed in this paper (No. 8, 9, and 13), and the items derived from the actual live concert experience (No. 2, 3, 10, 11, 12, 15, and 16) and important points as stage costume design (No. 1, 4, 5, 6, 7, and 14). Moreover, similar to conventional design guidelines, aesthetics are essential. The position and size of the control-device are related to ease of dance and comfort. Also, with the conventional design guidelines, wearable computing is mostly assumed for daily life or sports [28–31]. With our design guidelines, however, LED costumes are assumed for dance. In addition, proxemics and long-term use are in conventional design guidelines. Although it is better for a dancer to be able to wear an LED costume for a long time without him/her noticing the electronic equipment, many LEDs are attached to a costume and it is impossible not to perceive all LEDs. Since LED costume is only worn during the performance, the design guidelines of proxemics and long-term use for dancing are less restricted than that for daily use. The control-device should not be in a position where the dancers can touch it. When some problems occur with the control-device, the staff can repair it.

## 8. New LED Costume Along Our Design Guidelines

We fabricated a new LED costume based on the design guidelines proposed in Section 7. This costume is shown in Figure 9. It uses 1180 LEDs, consisting of 2 types of LED units and ring-shaped LED units. One uses addressable LED strip that consists of 11 LEDs. The LED source is a 5050SMD built-in SK6812 chip, and PCB thickness is 0.25 mm. The other uses addressable LED strip that consists of 6 LEDs. The LED source is the same as the former. The ring-shaped LED unit source is 5050SMD built-in SK7812 chip, and PCB thickness is 1 mm. The inner diameter is 30 mm, and outer diameter is 45 mm. Each LED unit can be connected with a JST SM connector. There are 94 LED units, which is more than for the previous LED costumes we fabricated. The LED costume fabricated in 2013 used 500

LEDs and 32 LED units, the LED costume fabricated in 2014 used 854 LEDs and 60 LED units, and the LED costume fabricated in 2015 used 867 LEDS and 32 LED units. Thus, the wiring of the new LED costume is more complex. The main fabric is nylon that is soft, light, and quick-drying, and the inner mesh is attached to wear easily without the cable and connector touching the wearer's skin. The inner mesh is also washable. The pants are integrated with the skirt on which the LEDs are attached. The LED units around the thigh and hip that can easily break are arranged so as to not bend. Placing LED units around the joints is avoided as much as possible. Ring-shaped LED units are used instead. A ring-shaped LED unit has a hard circuit board; thus, it is difficult to break. The fastener tape is sewn onto the surface of the costume, and the buttonholes are opened at the ends of the fastener tape. This enables fixing an LED unit with fastener tape and hiding the wiring cable into the costume through the buttonholes. Ring-shaped LEDs are only sewn by hand. Since the LED units are fixed with fastener tape, a broken LED unit can be replaced immediately. Since the number of sewing processes is less than conventional LED costumes, production costs can also be reduced. Moreover, 2 types of LED units are used and they are easy to remove from the costume, and it takes approximately 15 min to install LEDs, which is shorter than the conventional LED costume. A polycarbonate plate is attached to the back of all LED units to prevent an LED unit from creasing. The handcrafted reinforcement of the LED unit is shown in Figure 10. By pasting the polycarbonate plate between the LED strip and fastener tape with glue and fixing them with plastic tape at the ends, the LED strip is reinforced. Polycarbonate is safe because it is difficult to break. It can also return to its original shape after being bent, although it is relatively flexible; thus, it can be bent according to the movement of the dancer and cannot make a crease. We investigated two reinforcement materials. One is an acrylic plate, and the other is a polycarbonate plate. We investigated 3 acrylic plates of thicknesses 2 mm, 3 mm, and 5 mm. The 2 mm and 3 mm thickness plates are too soft to reinforce an LED strip. Moreover, the 5 mm thickness acrylic plate cracks when bent too much. If it breaks, it is possible that the debris hurt the dancer, therefore it is dangerous to use it for costumes. On the other hand, we investigated the GP polycarbonate plates [38] of thicknesses 2 mm, 3 mm, and 5 mm. The 3 mm and 5 mm thickness plates are too hard to bend, which disturbs dancing. The 2 mm thickness plate bends moderately and does not interfere with dancing. Therefore, a polycarbonate plate of thickness is 2 mm is selected.

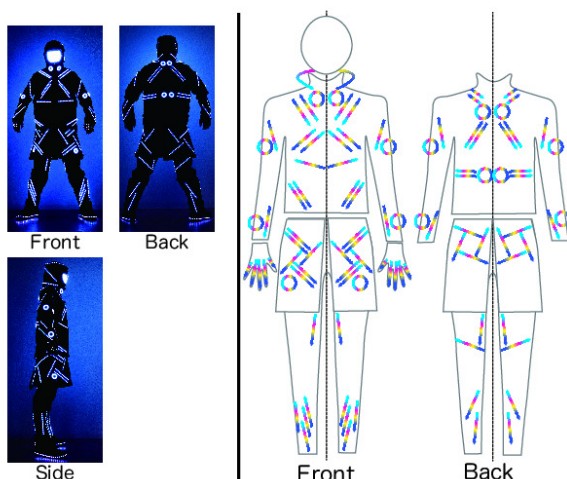

**Figure 9.** New LED costume along our design guidelines.

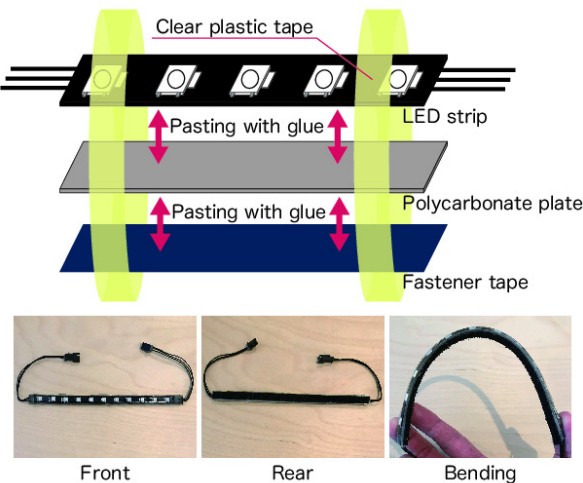

**Figure 10.** Handcrafted reinforcement of LED units.

## 9. Endurance Test Involving Dancing

Once LEDs are broken, the quality of the dance performance is deteriorated, and the staff have to spend effort on maintenance. Although the proposed design guidelines, except for making the LED hard to break, are important, fabricating an unbreakable LED costume is a major factor in improving quality overall. Therefore, we conducted an endurance test for the new LED costume created that involved actual dancing to verify the usefulness of the proposed design guidelines.

*9.1. Procedure*

We fabricated an experimental costume that has the new LED layout mentioned in Section 8 on the left half, and the older LED layout that was created in 2014 on the right half. The dancer wore this costume and danced freely. The experimental costume is shown in Figure 11. The costume fabricated in 2015 is the latest costume (Figure 2), which is a jumpsuit. Thus, in this endurance test, the LED layout created in 2014 was adopted due to use the same type of costume along with our proposed design guideline. When dancing, the observer turned on the LEDs every five minutes to check for failures. If a failure occurred, the position of the broken LED was recorded. After repairing it, the experiments restarted. The dancers have been dancing for 10 years (180 cm in height and weighing 68 kg) and have worked as a dancer for well-known artists in the past and have experience wearing LED costumes. This experiment was conducted for 1 h a day, for a total of 10 h for 10 days. The past concerts we participated in were held more than 20 times. Therefore, this endurance test time is equivalent to 20 concerts involving 30-min dance performances.

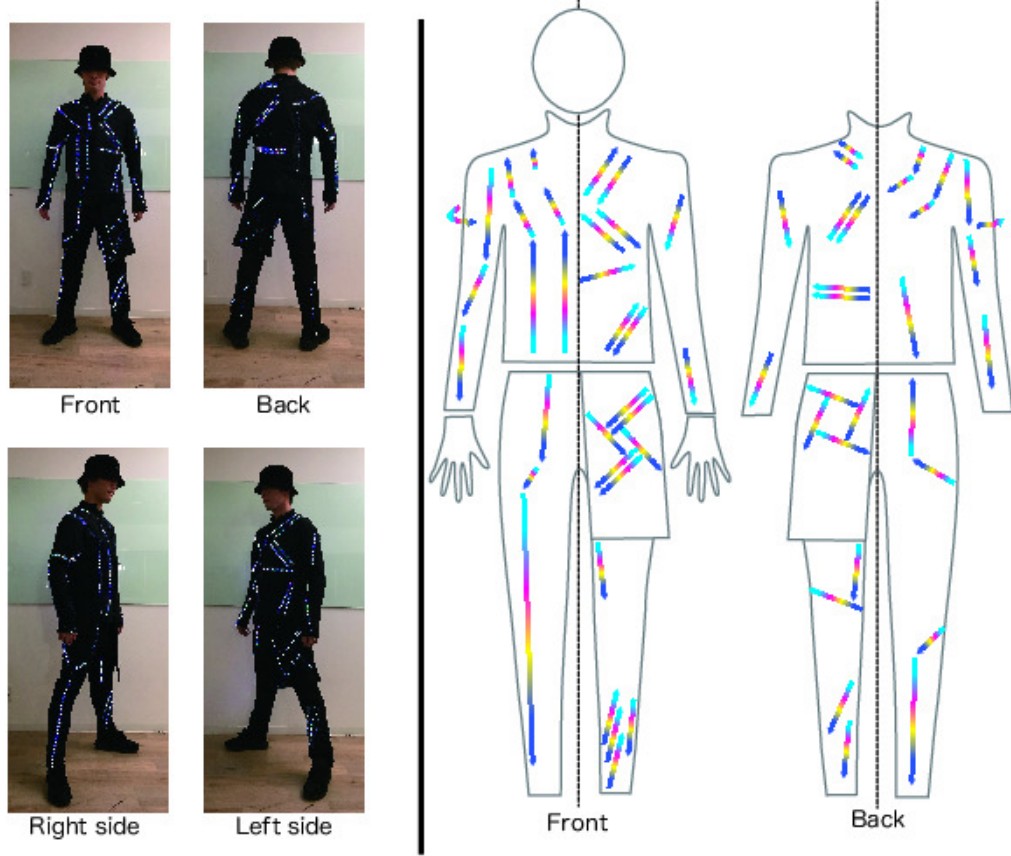

**Figure 11.** Experimental LED costume for endurance test.

*9.2. Result*

The results of this endurance test are shown in Figure 12. The notation $X^N$ symbol denotes the broken LED position, and $N$ denotes the number of broken LEDs in this test. The LED units located on the right elbow broke eight times. The LED units on the side of joints easily broke, the same as during the endurance test mentioned in Section 6. The dancer bent and stretched his elbow repeatedly while dancing, which caused fractures. Also, the two LED units on the lower half of the body broke. Approximately 5 to 10 min before these LED units broke, the dancer sat on the floor holding his knees, as shown in Figure 13. Thus, this posture is related to the LED breakage located on the lower body. From the analysis mentioned in Section 5, we found that the LED units around the thigh break more frequently than those around the rest of the body. However, there is a possibility that they were not broken by dance. Dancers usually sit on a chair or on the floor until performance time after wearing the LED costume. From the results in Table 4, the movement of repeatedly sitting and standing can easily break the LED units located around the groin. Therefore, waiting postures, such as sitting, are considered more damaging to LEDs than dancing. We hence add to the proposed design guidelines.

17. Waiting postures: Dancers must wait for their turn for a long time while wearing LED costumes. Therefore, the design of LED costumes must take into account the postures during waiting time.

From the observation of dancing, the elbows were bent significantly more than the knees and around thigh and hip. It is considered that the LED units located on the upper half of the body, such as the elbows, break due to dancing, and the LED units located on the lower half of the body break due to the waiting postures such as sitting on a chair or the floor. On the other hand, the LED layout on the left side of the body, which is based on our proposed design guidelines, did not break during 10 h of dancing. Therefore, proper LED layout and reinforcement can decrease LED breakage.

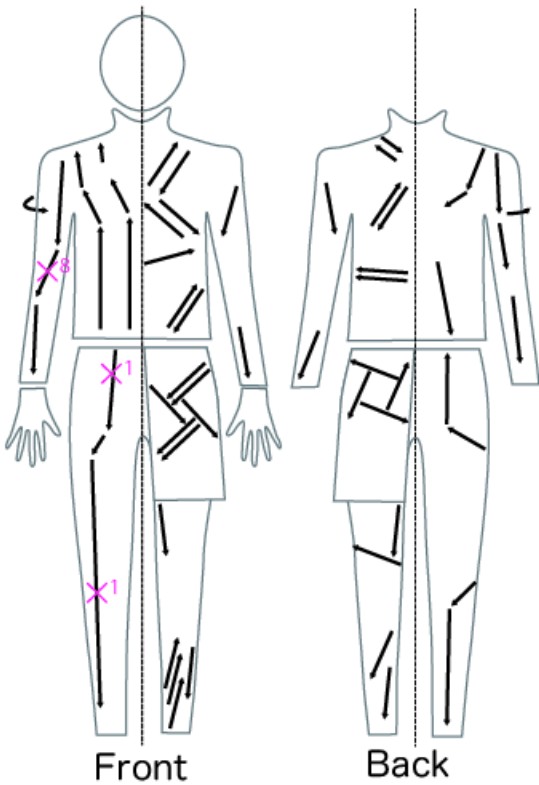

**Figure 12.** Results from dance endurance test.

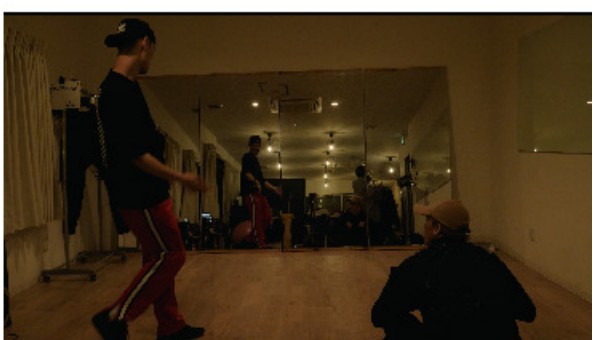

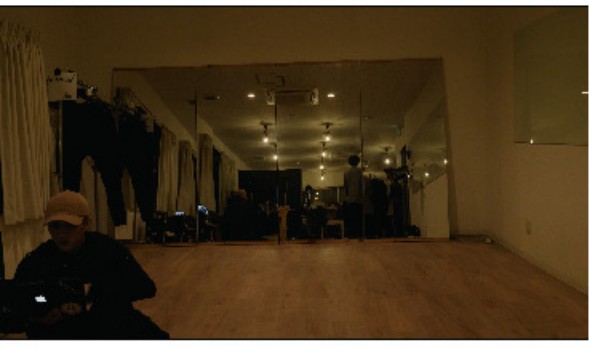

**Figure 13.** Sitting postures of dancer.

## 10. Dance Performance with New LED Costume

We conducted two dance performances using the LED costume base on our design guidelines introduced in section 8. These dance performances are shown in Figure 14. In the Digital Contents EXPO 2018 [39], held in Japan in October, we conducted a special dance performance that imitates fencing jointly with Dai Nippon Printing. There were two dancers, and they wore an LED mask and held an LED sword reminiscent of fencing. There were one or two technical staff members including the operator. This performance time was approximately 3 min and conducted 3 times per day for 3 days, and the LEDs on the costumes did not break. At the OSAKA AUTO MESSE 2019 [40] and TOKYO AUTO SALON 2019 [41], Japanese car exhibitions held in January and February 2019, respectively, we conducted special dance performances as a part of the advertising of a new sports car at the TOYOTA GAZO RACING booth. There were ten dancers, and the performance included by a powerful dance with a sense of running. There were two or three technical staff members including the operator. The performance time was approximately 3 min, and was performed 4 or 5 times per day for 6 days. Some LED units got fractures because of the powerful dance. Since only a few LED units were fractured per day, they could be repaired by one staff member. The main cause of failure was that the LED units peeled off from the polycarbonate plate. From these two performances, the number of broken LED has decreased. Due to the little breakage, we could reduce the number of staff members to repair the broken LEDs and maintain the costume, reducing cost.

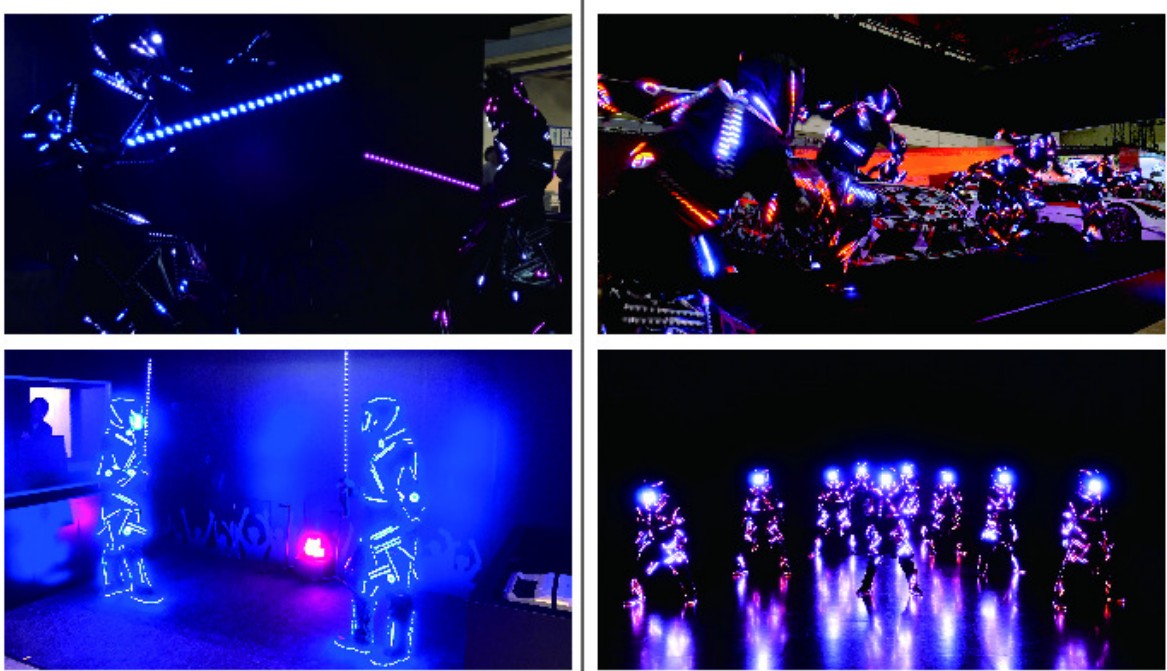

**Figure 14. Left**: fencing special dance performance **Right**: special dance performance at motor shows.

## 11. Conclusions

### 11.1. Conclusions

We proposed design guidelines on LED costumes for dance performances assuming repetitive use, based on our experience of using LED costumes at large concert tours for 5 years and the endurance tests mentioned in this paper. We fabricated a new LED costume based on our design guidelines. From the endurance test involving dance, we found that the LED layout based on our proposed guidelines and handcrafted reinforcement reduced LED failures. Also, from the two endurance tests, we found that the cause of LED failure is not only dance but also other factors, such as the postures of the dancers

during waiting time. For future work, we will conduct further research on the factors causing LED failure, and update our design guidelines.

*11.2. Contribution*

Garments are often closely related to a particular industry. For example, from 1853 to 1873, Levi Strauss, who immigrated from Germany to the United States during the gold rush, developed durable, easy to move in, and convenient work pants for people working in the gold mines. His pants spread throughout the world as *jeans*. It is no exaggeration to say that the jeans were produced by the coal-mining industry. Also, hunting jackets have leather reinforcements called a *gun patch* sewn onto the shoulders and elbows to prevent abrasion. This has been incorporated into various types of clothing as design. This is considered to have sublimed into the design a small idea that originated from the hunting industry. A computer that occupied more than half of a desk just 10 years ago can now fit in a pocket and has a more powerful processor. The LED costume is a new genre and style generated from the development of information technology. It is necessary to have both knowledge of clothing and wearable computing to establish the style of the LED costumes. In this paper, we proposed design guidelines on LED costumes that take into account clothing and wearable computing, and fabricated a new LED costume based on these guidelines. If the costs of fabrication and operation decrease, LED costumes will become indispensable for live performances. The dance performances using LED costumes targeted in this study are one of the harshest environments for using wearable computing. Although wearable computing during a live performance is not used often due to stability issues and high cost, it will spread because of the high quality of performance. Our strong point is that LED costumes are made by ourselves and we use them at large concerts throughout the year. We believe that the knowledge and data obtained from this study can greatly contribute to the future development of wearable computing for live entertainment. Furthermore, the wiring for an LED costume is applied to biosensing. Wearable sensors must have more precise wiring considered for accurate sensing and durability than that of LEDs. Therefore, the data collected in this study and our proposed design guidelines can contribute to other fields such as sports and medicine.

Although stage-costume design is important, we are not merely dealing with the consumption of design and fashion technology, but are exploring LED costume styles that can enable the best performance at low cost. We hope that our design guidelines and LED costume design will liven up the live-entertainment industry and be handed down to future generations.

**Author Contributions:** Writing—original draft, Formal analysis, R.I.; Writing—review & editing, T.T.; Supervision, Y.Y., M.T.; Conceptualization, M.F.

**Funding:** This work was supported by JST CREST Grant Number JPMJCR18A3, Japan.

**Conflicts of Interest:** The authors declare no conflicts of interest.

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
