# Peer review of "Design Guidelines on LED Costumes for Dance Performances"

_designs_

Round 1
Reviewer 1 Report
The paper aims to give details of improvements to the design of dance costumes with attached LED lighting units. This is a significant and original. Extensive literature and performance review, plus several experiments are described. These are useful in leading to conclusions about optimal design for costumes covered in LEDs. The material is there, but requires extensive revision to make a clear, concise paper where the overall conclusions are made based on experiments described in the results section.
The authors have presented a good range of the associated literature on performance and technology, but require some more references. The way in which the details of other costumes and performances lead to the authors’ own, innovative costume designs needs to be made clear.
The questions that the researchers are asking and solving need to be made clearer. The paper refers to the use of LED lighting units, not discrete LEDs. This needs to be made clear in the title and throughout the paper. Details of the LED units used, plus associated hardware should be included within the paper. Details of the overall costume design, such as wiring, batteries and their placement, are lacking. There is a general lack of scientific soundness, for example LED units failed due to a dancer sitting down during tests. Useful design guidelines ‘appear’ in section 7 without previous mention in the experimental section. Discussions with a scientist or engineer accustomed to scientific publishing could be useful when revising the paper.
Extensive work has obviously gone into the making of the costumes described in this paper. There are very useful lessons that the authors are sharing as a result of this. A clearer, revised paper will be a useful contribution to the fields of wearable technology and performance.
Details by line number are given below:
15 – 32 Addition of references would strengthen the introduction.
34 A reference is needed to a costume with mesh pockets for storing LEDs. How does this make it easier to maintain the costume, and if this is so, why are further developments required?
44 This statement needs clear follow up later in the paper.
59 how does the retroreflective cloth make it easier to identify skin and ‘other space’?
120 Reference needed
123 Give details of this costume and, if possible, a reference.
139 Explain why this was problematic.
144 lit not slit
148 provide a link or reference to the performance video, if possible
159 reference needed
172 How did the lighting enhance the dance, not simply light up the costume? The reader cannot tell from the photo in Figure 1.
183 Reference needed to back up this statement. Note that laser lighting can be stronger than LED lighting.
193 A reference to back up this statement would be good. What is the limitation?
203 are not mare
249 What is meant by a 'proper position'?
266 Best to add some detail of the different demands that a dance performance places on a costume.
273 Repetition of the need to keep secrets. Best to leave this information out and present the work that can be published without causing IP issues.
283 Future work: cover movement and storage of costumes.
293 Give some overall description of the costume before going into details of separate parts. Describe what changes were made for each update, and how this affected the LED failure rate.
295 Is the wiring bent and stretched repeatedly not the LEDs themselves?
Figure 2: What is meant by ‘ring-shaped’ LED? This is several individual LEDs in a ring structure. Precise details of this part are required.
300 Give full details of the LED strips, including manufacturer.
307 Give precise details of the lengths of the LED strips: longer and shorter ones.
336 Do you mean LEDs or LED units that broke?
351 How do you know that the heat and sweat of the dancer caused this type of breakage?
352 What is meant by rending? More detail is needed about the way in which the cable got caught?
Figure 3. Make the caption clearer: This is the front, rear, side of the LED strip, not the positions on the dancer as in Figure 4.
360 The costume is symmetrical, but dance rarely is! Better to say that only one side of the costume was assessed due to ?time constraints. This is a limitation to the study.
371 Was the endurance test based on any previous work, or on standards or guidelines?
Figure 5: More detail in the caption will help the reader to understand how the diagrams on the left relate to the photos on the right.
392 Why were different amounts of repetitions of the test chosen for different body parts?
Tables 2 – 4: It would be easier to understand how these related to the photos in Figure 5 if there were some more photos or diagrams alongside the tables.
406 Explain how the finding about LED unit fracture relates to the points that are in the tables (Af etc).
416 – 462 These are important findings, but are different from the material presented in the results. Statemetns such as ‘We selected it based on our experience’ are vague. More detail is needed, and ideally the way in which these important points were discovered should be included within the method and results or put into another paper.
477 How is the ‘more complex’ wiring constructed and arranged on the costume?
482 It is unclear what the buttonholes are for.
489 How does the polycarbonate affect the dancers?
550 How did LEDs break ‘a little’?
Author Response
The paper aims to give details of improvements to the design of dance costumes with attached LED lighting units. This is a significant and original. Extensive literature and performance review, plus several experiments are described. These are useful in leading to conclusions about optimal design for costumes covered in LEDs. The material is there, but requires extensive revision to make a clear, concise paper where the overall conclusions are made based on experiments described in the results section.
The authors have presented a good range of the associated literature on performance and technology, but require some more references. The way in which the details of other costumes and performances lead to the authors’ own, innovative costume designs needs to be made clear.
The questions that the researchers are asking and solving need to be made clearer. The paper refers to the use of LED lighting units, not discrete LEDs. This needs to be made clear in the title and throughout the paper. Details of the LED units used, plus associated hardware should be included within the paper. Details of the overall costume design, such as wiring, batteries and their placement, are lacking. There is a general lack of scientific soundness, for example LED units failed due to a dancer sitting down during tests. Useful design guidelines ‘appear’ in section 7 without previous mention in the experimental section. Discussions with a scientist or engineer accustomed to scientific publishing could be useful when revising the paper.
Extensive work has obviously gone into the making of the costumes described in this paper. There are very useful lessons that the authors are sharing as a result of this. A clearer, revised paper will be a useful contribution to the fields of wearable technology and performance.
Details by line number are given below:
15 – 32 Addition of references would strengthen the introduction.
Response 1: We add references.
34 A reference is needed to a costume with mesh pockets for storing LEDs. How does this make it easier to maintain the costume, and if this is so, why are further developments required?
Response 2: If there is no mesh pocket, the LED must be sewn to the garment, and if the LED breaks down, it is necessary to re-sew it, which is labor intensive. In addition, costumes with mesh pockets make easier to store LEDs, but wiring are inside the costume to conceal, and if the LED breaks, the dancer must wear off the costume for repair.
44 This statement needs clear follow up later in the paper.
Response 3: The design guidelines for wearability are mainly intended for daily use as described in section 3. Based on this, we have proposed a design guideline for LED costumes specialized in dance performance in Chapter 7.
59 how does the retroreflective cloth make it easier to identify skin and ‘other space’?
Response 4:The retroreflective material has the property of reflecting light in the direction of the light source, and when it is irradiated with infrared light, the reflection can be photographed with an infrared camera to obtain high brightness. Therefore, costumes, skin, and other spaces are identified from the difference in brightness of the reflection.
120 Reference needed
Response 5:We add the reference.
123 Give details of this costume and, if possible, a reference.
Response 6: The detail of this costume can be seen in reference[14]
139 Explain why this was problematic.
Response 7: If there are LEDs on the joints such as elbows and knees, the LEDs are bent and stretched frequently, so there is a high possibility that the LEDs will break and break down.
144 lit not slit
Response 8: We fix it.
148 provide a link or reference to the performance video, if possible
Response 9: We add link to the performance video
159 reference needed
Response 10: The detail of the research can be seen at reference [23].
172 How did the lighting enhance the dance, not simply light up the costume? The reader cannot tell from the photo in Figure 1.
Response 11: By controlling the lighting timing of the LED to the music, for example, only arms other than the main dancer is lit, and they are arranged in a line, a dancer can dance as if the dancer have many arms.Thus, we are aiming for dances that are expanded by light in this way.
183 Reference needed to back up this statement. Note that laser lighting can be stronger than LED lighting.
Response 12: The light intensity of the light emitting device is expressed numerically, but unlike the light of the LED, the light of the laser light cannot be seen directly, so it feels a little dark.
193 A reference to back up this statement would be good. What is the limitation?
Response 13: The constraints are the accuracy of person tracking and the number of dancers that can project at one time.
203 are not mare
Response 14: We fix it.
249 What is meant by a 'proper position'?
Response 15: By attaching a sensor to the body as described in the reference [30], it is possible to acquire body movement data, but the position of the sensor must be determined in consideration of the body movement.
266 Best to add some detail of the different demands that a dance performance places on a costume.
Response 16: Based on the our experiment of the use of LED costumes, the requirements that dance performance imposes on costumes are described in Section 7.
273 Repetition of the need to keep secrets. Best to leave this information out and present the work that can be published without causing IP issues.
Response 17: The costume shown in figure 2 cannot be posted as a whole due to confidentiality. Therefore, only partial parts are mentioned in the illustration.
283 Future work: cover movement and storage of costumes.
293 Give some overall description of the costume before going into details of separate parts. Describe what changes were made for each update, and how this affected the LED failure rate.
Response 18: We add overall description of the costumes.
295 Is the wiring bent and stretched repeatedly not the LEDs themselves?
Response 19: No, LED strip is bent and stretched.
Figure 2: What is meant by ‘ring-shaped’ LED? This is several individual LEDs in a ring structure. Precise details of this part are required.
Response 20: it means ring-shaped LED strip. It consists of 16 LEDs and is arranged in a ring shape.
300 Give full details of the LED strips, including manufacturer.
Response 21: We add the detail of the LED strip.
307 Give precise details of the lengths of the LED strips: longer and shorter ones.
Response 22:We add the length of the LED strips.
336 Do you mean LEDs or LED units that broke?
Response 23: Yes, we collected broken LED on constumes during live concerts.
351 How do you know that the heat and sweat of the dancer caused this type of breakage?
Response 24: These LEDs had water drops between the parts, and were often rusted.
352 What is meant by rending? More detail is needed about the way in which the cable got caught?
Response 25: In this paper, "rending" means "tear-off". Each LED unit is connected with a cable, but if the cable is entangled or does not fit in the specified location, it will be tear-off during the dance.
Figure 3. Make the caption clearer: This is the front, rear, side of the LED strip, not the positions on the dancer as in Figure 4.
Response 26: Figure 3 is an explanation of the appearance of our original LED itself, and shows that the front is filled with silicone resin.
360 The costume is symmetrical, but dance rarely is! Better to say that only one side of the costume was assessed due to ?time constraints. This is a limitation to the study.
Response 27: Depending on the dance, there is a possibility that the degree of breakage on both side is different. However, not the influence of the dance, but the influence of the design of the costume itself is being investigated.
371 Was the endurance test based on any previous work, or on standards or guidelines?
Response 28: Although the actual endurance experiment is performed mechanically, this time, because it is assumed to be used for dance performance, an experiment reflects the movement of the body.
Figure 5: More detail in the caption will help the reader to understand how the diagrams on the left relate to the photos on the right.
Response 29: we add detailed caption.
392 Why were different amounts of repetitions of the test chosen for different body parts?
Response 30:The damaged LED was clearly identified at the beginning of the experiment. Therefore, this is the number of times that the damaged LED was broken.
Tables 2 – 4: It would be easier to understand how these related to the photos in Figure 5 if there were some more photos or diagrams alongside the tables.
Response 31: If a figure is placed next to the table, the figure becomes smaller and the characters in the figure become difficult to read.
406 Explain how the finding about LED unit fracture relates to the points that are in the tables (Af etc).
Response 32: In this test, we found that the LED unit located at the side of the joints and around the groin easily.
416 – 462 These are important findings, but are different from the material presented in the results. Statemetns such as ‘We selected it based on our experience’ are vague. More detail is needed, and ideally the way in which these important points were discovered should be included within the method and results or put into another paper.
Response33: The design guidelines proposed in this paper are the items derived from the analysis and endurance experiments performed in this paper (No. 8, 9, and 13), and the items derived from the actual live concert experience (No. 2, 3, 10, 11, 12, 15, and 16) and important points (No. 1, 4, 5, 6, 7, and 14) as stage costume design. In order to verify the proposed design guidelines, it is necessary to use at live concerts. Therefore, it is as the future.
477 How is the ‘more complex’ wiring constructed and arranged on the costume?
Response 34: This sentence means that new LED costume uses more LED units than the previous LED costume descripbed in section 4. Therefore, we remove the sentence.
482 It is unclear what the buttonholes are for.
Response 35: This is to pass the cable inside the costume.
489 How does the polycarbonate affect the dancers?
Response 36: A dancer feel the presence of LEDs, but does not affect dance.
550 How did LEDs break ‘a little’?
Response 37: LED strips were fractured.
Reviewer 2 Report
This is a very interesting paper that develops design guidelines on light emitting diode (LED) costumes for dance performances for repetitive use. Based on extensive testing of the LED costume design in real-life concerts, the authors devised 16 design guidelines for LED costumes in dance performances and produced an improved LED costume based on these design guidelines. Subsequent endurance tests demonstrated that the LEDs no not break when they are repetitively used.
A key notion in this study is the so-called "design guideline". You're using this notion somewhat loosely, although previous work in the area of design research implies one should formulate "design principles" more systematically. For example, the following articles show how to formulate a design principle in terms of Contexts, Interventions, generative Mechanisms and Outcomes (CIMO):
Denyer, D., Tranfield, D. & Van Aken, J. E. (2008). Developing design propositions through research synthesis. Organization Studies, 29(3), 393-413. Pascal, A., Thomas, C., & Romme, A. G. L. (2013). Developing a human-centred and science-based approach to design: The knowledge management platform project. British Journal of Management, 24(2), 264-280. Van Burg, E. & Romme, A. G. L. (2014). Creating the future together: Toward a framework for research synthesis in entrepreneurship. Entrepreneurship Theory and Practice, 38(2), 369-397.Therefore, I strongly recommend you use a CIMO format for the design guidelines/principles regarding LED costume design. By means of this format, you force yourself to uncover any (otherwise implicit) elements, such as the specific contextual setting in which the LED costume design can be effectively used as well as the underling theoretical mechanisms. A systematic (CIMO-like) format of your design principles also make them more likely to be replicated and/or extended in future work.
A key deficiency of the paper is the English writing. In almost every sentence the first-person "we" is used, which undermines the readability and flow of the text. As non-native authors, you therefore need the help from a native writer to rewrite and polish the entire text. The current writing style also leads to possible confusion, for example, in the sentence starting with "As concert personnel, we are obliged to (...)" (lines 26-27 in manuscript). I cannot imagine the authors were actually operating as concert personnel! I would thus rewrite this sentence, for instance, as follows:
In testing out the LED costume design in actual concerts, the authors of this article needed to conform to the non-disclosure conditions in ongoing concerts.
All the best in revising this manuscript and getting it published!
Author Response
This is a very interesting paper that develops design guidelines on light emitting diode (LED) costumes for dance performances for repetitive use. Based on extensive testing of the LED costume design in real-life concerts, the authors devised 16 design guidelines for LED costumes in dance performances and produced an improved LED costume based on these design guidelines. Subsequent endurance tests demonstrated that the LEDs no not break when they are repetitively used.
A key notion in this study is the so-called "design guideline". You're using this notion somewhat loosely, although previous work in the area of design research implies one should formulate "design principles" more systematically. For example, the following articles show how to formulate a design principle in terms of Contexts, Interventions, generative Mechanisms and Outcomes (CIMO):
Denyer, D., Tranfield, D. & Van Aken, J. E. (2008). Developing design propositions through research synthesis. Organization Studies, 29(3), 393-413. Pascal, A., Thomas, C., & Romme, A. G. L. (2013). Developing a human-centred and science-based approach to design: The knowledge management platform project. British Journal of Management, 24(2), 264-280. Van Burg, E. & Romme, A. G. L. (2014). Creating the future together: Toward a framework for research synthesis in entrepreneurship. Entrepreneurship Theory and Practice, 38(2), 369-397.
Therefore, I strongly recommend you use a CIMO format for the design guidelines/principles regarding LED costume design. By means of this format, you force yourself to uncover any (otherwise implicit) elements, such as the specific contextual setting in which the LED costume design can be effectively used as well as the underling theoretical mechanisms. A systematic (CIMO-like) format of your design principles also make them more likely to be replicated and/or extended in future work.
Response 1: Thank you for your recommendation. However, since this research does not fit well in CIMO-logic format, we described to enumerate the design guidelines.
A key deficiency of the paper is the English writing. In almost every sentence the first-person "we" is used, which undermines the readability and flow of the text. As non-native authors, you therefore need the help from a native writer to rewrite and polish the entire text. The current writing style also leads to possible confusion, for example, in the sentence starting with "As concert personnel, we are obliged to (...)" (lines 26-27 in manuscript). I cannot imagine the authors were actually operating as concert personnel! I would thus rewrite this sentence, for instance, as follows:
In testing out the LED costume design in actual concerts, the authors of this article needed to conform to the non-disclosure conditions in ongoing concerts.
Response 2: We reduce the use of the first person in writing, and fix the sentence (lines 26-27 in manuscript).
All the best in revising this manuscript and getting it published!
Reviewer 3 Report
The subject is interesting and closely related to current actual demand. It is valuable for the future LED costumes industry to refer from the design principles derived from practical applications. However, there are several shortages that jeopardized the value of the study, which are listed below.
1.The use of the first person should be reduced in writing. Using the first person writing style should be mainly in Introduction and Discussion sections. The writing style of using the first person throughout this paper must be improved, otherwise this writing style can lead to very subjective research.
2. The structure is quit long. However, the relationship between each section should be clearer. It is suggested that the authors should provide a figure of the research structure.
3. The study was conducted mainly on LED costumes. So there is no need for such a lengthy discussion of other lighting costume technologies in section 2. Lighting costume technology. Authors should consider make this section shorter and more focused.
4. What is the meaning of 4. Our achievement?This subtitle is amateurish. It doesn't matter what you have achieved in this part of the manuscript (before methods), it's how the achievement was made scientifically and problems identified in previous studies.
5. Section 7. Design Guidelines for LED costume should be the most valuable section in the study. However, the authors should compare the new design principles with those of previous studies. Without the comparison the new design guidelines for LED costume now seem subjective.
Author Response
1.The use of the first person should be reduced in writing. Using the first person writing style should be mainly in Introduction and Discussion sections. The writing style of using the first person throughout this paper must be improved, otherwise this writing style can lead to very subjective research.
Response 1: We reduce the use of the first person in writing.
2. The structure is quit long. However, the relationship between each section should be clearer. It is suggested that the authors should provide a figure of the research structure.
Response 2: At the end of the introduction and the beginning of each chapter, we explain the relationship between each chapter.
3. The study was conducted mainly on LED costumes. So there is no need for such a lengthy discussion of other lighting costume technologies in section 2. Lighting costume technology. Authors should consider make this section shorter and more focused.
Response 3: Since the costume technologies other than LED costumes are also important as reference materials, we keep them.
4. What is the meaning of 4. Our achievement?This subtitle is amateurish. It doesn't matter what you have achieved in this part of the manuscript (before methods), it's how the achievement was made scientifically and problems identified in previous studies.
Response 4: We used LED costumes more than 100 performances per year and have various knowledge about the use of LED costumes. Section 4 shows that we have improved LED costumes using our knowledge. This is indispensable when proposing design guidelines. This section is to prove that we are an organization worthy of proposing design guidelines. We change the title of section 4 to avoid misunderstanding.
5. Section 7. Design Guidelines for LED costume should be the most valuable section in the study. However, the authors should compare the new design principles with those of previous studies. Without the comparison the new design guidelines for LED costume now seem subjective.
Response 5: As far as we know, there is no design guideline specialized in the fabrication of LED costumes. In addition, the design guidelines for wearability described in section 3 have a relatively close relationship with that of LED costumes. However, since they are mainly intended for daily use or sports, it seems difficult to compare them equally as described from line 455.
Reviewer 4 Report
I found the paper innovative and interesting. Overall the manuscript is excellent about the 『Design Guidelines on LED Costumes for Dance Performances』. However, the authors shall provide further analysis model or evidence to verify the guideline is effectively and efficiency while setup design guideline is in Section 7. Moreover, the authors shall not state the 『We selected it based on our experience』 in the science journal whereas the reviewer suggests it to be removed in the paper before the publish.
Author Response
I found the paper innovative and interesting. Overall the manuscript is excellent about the 『Design Guidelines on LED Costumes for Dance Performances』. However, the authors shall provide further analysis model or evidence to verify the guideline is effectively and efficiency while setup design guideline is in Section 7.
Response 1: Based on the proposed design guidelines, we fabricated the LED costume in section 8, and conducted the endurance test involving dancing in section 9. It focuses on LED failures. Therefore, the endurance test in section 9 shows how important it is to fabricate LED costumes considering the (8)LED layout constraints and (13)Moderately flexible LED strip listed in the proposed design guidelines. Also, (10)drip-proof and water proof, (11)adjustable, and (15)Washable are proposed based on our actual experiences of using LED costumes at concerts, and it is clear that they should be considered when fabricating LED costumes. Verification of the other design guidelines requires multiple patterns of costumes, which must be used in the same environment. It takes time and money. Therefore, it will be verified as future works.
Moreover, the authors shall not state the 『We selected it based on our experience』 in the science journal whereas the reviewer suggests it to be removed in the paper before the publish.
Response 2: We remove the sentence.
Round 2
Reviewer 1 Report
This paper still requires extensive revision to make a clear, concise paper where the overall conclusions are made based on experiments described in the results section. The authors have only addressed a few of the points that I made.
The paper refers to the use of LED lighting units, not discrete LEDs. This still needs to be made clear in the title and throughout the paper. The authors are unwilling to give details of the LED units used, plus associated hardware. Generic details of the overall costume design, such as wiring, batteries and their placement, are lacking. Generic details must be included so that the reader is aware what is included on the costume. Without this, the paper lacks sufficient scientific rigour for publication. If the authors cannot include details, then the paper should be turned into a literature review with minimal details of their own costumes.
Extensive work has obviously gone into the making of the costumes described in this paper. There are very useful lessons that the authors are sharing as a result of this. A clearer, revised paper will be a useful contribution to the fields of wearable technology and performance.
Details by line number are given below. Note that the numbering refers to the lines in version 1, where a change or clarification was requested. I have either written ‘Y’ after the original comment, to say that it has been successfully addressed in version 2 of the paper, or I have written V2: followed by details of further information that is required. The authors have addressed very few of my original comments. In particular, no details are given of the LED lighting units and wiring plus control systems that were used. This renders the study useless, as the reader does not know the details of the technology that is being discussed. It is possible to give broad details of this technical information without compromising the confidentiality of other aspects of the design, such as who the costumes were made for and where they were used.
15 – 32 Addition of references would strengthen the introduction. Y
34 A reference is needed to a costume with mesh pockets for storing LEDs. How does this make it easier to maintain the costume, and if this is so, why are further developments required? V2: The reference does not explain the mesh pockets.
44 This statement needs clear follow up later in the paper. V2: where has this been added?
59 how does the retroreflective cloth make it easier to identify skin and ‘other space’? V2: No explanation has been added.
120 Reference needed Y
123 Give details of this costume and, if possible, a reference. V2: No details added
139 Explain why this was problematic. V2: No explanation added.
144 lit not slit. Y (word changed)
148 provide a link or reference to the performance video, if possible. V2: No link provided
159 reference needed. Y (I have now noticed that the reference was there in version 1)
172 How did the lighting enhance the dance, not simply light up the costume? The reader cannot tell from the photo in Figure 1. V2: no explanation provided.
183 Reference needed to back up this statement. Note that laser lighting can be stronger than LED lighting. V2: This section needs to be made clearer, for example why is light halation caused by LEDs, but not by EL wire? Is there a reference to support this?
193 A reference to back up this statement would be good. What is the limitation? V2: No reference added.
203 are not mare. V2: Should read ‘most are one-off events’
249 What is meant by a 'proper position'? V2: An explanation has not been provided.
266 Best to add some detail of the different demands that a dance performance places on a costume. V2: Best to explain what happens to the costumes that is different between an entertainment environment and other environments.
273 Repetition of the need to keep secrets. Best to leave this information out and present the work that can be published without causing IP issues. V2: There is still unnecessary repetition of the need to maintain IP. The basic details of the LED units and associated circuitry must be provided. The reader needs to know what technology was being used in order for the paper to be useful to anyone except the authors. There are very many different types of LEDs on the market, and they are placed together in lighting strips, lamps and many other types of assembly. Different LED units will respond differently to the stresses and strains of performance, so the paper needs to state which generic type of LED unit was used; what type of material the LEDs were placed in to hold the unit together; and how wiring was used to connect the units. For example are the LEDs held in a rubber or thermoset plastic substrate? Details such as these will make a great difference to the way in which the LED units break. Some detail is now supplied below Fig 2. Including this in a more comprehensive ‘materials’ or similar section at the start of the paper would make it easier for the reader to understand what was being tested. A scientific paper must contain information that will enable the reader to repeat the experiments that are described. Without this information, there is insufficient scientific rigour for the paper to be published. (It is not necessary to publish details of the performances.)
283 Future work: cover movement and storage of costumes. V2: This is not addressed. It was a suggestion for use in further work, but can be left out here.
293 Give some overall description of the costume before going into details of separate parts. Describe what changes were made for each update, and how this affected the LED failure rate. V2: Some detail of the costume is needed: for example was it always a body suit with LED units sewn or glued on? If so, what type of body suit made from what material? This information should be included in with the details of the LED units. A very basic diagram of the costume, showing what is included where, including battery placement, would enable the reader to understand what exactly is being discussed, and could be created without infringing IP.
295 Is the wiring bent and stretched repeatedly not the LEDs themselves? Y. The LED units themselves are bent and stretched repeatedly.
Figure 2: What is meant by ‘ring-shaped’ LED? Is this several individual LEDs in a ring structure? Precise details of this part are required. V2: no details supplied. The reader needs to know the basics: how many LEDs of what size are held together and connected how?
300 Give full details of the LED strips, including manufacturer. V2: See my comments for line 293. The reader needs to know the basic details of the type of technology being used, otherwise the paper is not scientifically sound. An ‘SM connector’ is now described in the new version of the paper. This needs a fuller explanation: what is an SM connector and where is it on the costume? How does it connect one unit to the next or to the battery? This information could be included without disclosing precise details, but a description of ‘LED units’ that are made of unknown materials and are held on a costume in unknown ways are insufficiently rigorous to be published.
307 Give precise details of the lengths of the LED strips: longer and shorter ones. V2 47 cm length now supplied. What were the dimensions of each LED unit used in the costume? How does 47 cm relate to the other units. Please add this to the generic details of the costume and LED units that need to be included in a ‘materials’ section.
336 Do you mean LEDs or LED units that broke? V2: Question not answered.
351 How do you know that the heat and sweat of the dancer caused this type of breakage? V2: Question not answered.
352 What is meant by rending? More detail is needed about the way in which the cable got caught? V2: No explanation supplied.
Figure 3. Make the caption clearer: This is the front, rear, side of the LED strip, not the positions on the dancer as in Figure 4. V2: Amendment not made to Fig. 3 caption.
360 The costume is symmetrical, but dance rarely is! Better to say that only one side of the costume was assessed due to ?time constraints. This is a limitation to the study. V2: no explanation supplied.
371 Was the endurance test based on any previous work, or on standards or guidelines? V2: no explanation supplied.
Figure 5: More detail in the caption will help the reader to understand how the diagrams on the left relate to the photos on the right. V2: No amendment supplied.
392 Why were different amounts of repetitions of the test chosen for different body parts? V2: No clarification supplied.
Tables 2 – 4: It would be easier to understand how these related to the photos in Figure 5 if there were some more photos or diagrams alongside the tables. V2: no diagrams have been supplied.
406 Explain how the finding about LED unit fracture relates to the points that are in the tables (Af etc). V2: No changes have been made.
416 – 462 These are important findings, but are different from the material presented in the results. Statements such as ‘We selected it based on our experience’ are vague. More detail is needed, and ideally the way in which these important points were discovered should be included within the method and results or put into another paper. V2: vague statements have been removed.
477 How is the ‘more complex’ wiring constructed and arranged on the costume? V2: Please make the change in the number of LED units clearer. How many LED units were on the previous version of the costume. Details of the costume are supplied in this section. They should also be in the materials and methods section, so that the reader knows what type of costume is being described early in the paper.
482 It is unclear what the buttonholes are for. V2: some explanation is now supplied.
489 How does the polycarbonate affect the dancers? V2: Insufficient detail given. ‘we investigated various reinforcement materials’ is too vague for a scientific paper. Which materials? How was the flexibility of polycarbonate assessed? The type of polycarbonate used needs to be stated, as the rigidity of polycarbonate is variable: https://www.acplasticsinc.com/informationcenter/r/acrylic-vs-polycarbonate
550 How did LEDs break ‘a little’? V2: This has been corrected. Thank you.
Author Response
34 A reference is needed to a costume with mesh pockets for storing LEDs. How does this make it easier to maintain the costume, and if this is so, why are further developments required? V2: The reference does not explain the mesh pockets.
Response 1: We added the explanation in line 35-39.
44 This statement needs clear follow up later in the paper. V2: where has this been added?
Response 2: This added in line 513-521.
59 how does the retroreflective cloth make it easier to identify skin and ‘other space’? V2: No explanation has been added.
Response 3: This has been added in line 68-71
123 Give details of this costume and, if possible, a reference. V2: No details added
Response 4: Such as CuteCircuit's costume[13] and sparkling crystal dress[14] fabricated by Chalayan, Costumes on which the LEDs are sewn have been fabricated over ten years ago.
(added in line 137-139)
139 Explain why this was problematic. V2: No explanation added.
Response 5: The explanation was added in line 154-155.
148 provide a link or reference to the performance video, if possible. V2: No link provided
Response 6: performance video link was add in line 165.
172 How did the lighting enhance the dance, not simply light up the costume? The reader cannot tell from the photo in Figure 1. V2: no explanation provided.
Response 7: we added the explanation in line 189-190.
183 Reference needed to back up this statement. Note that laser lighting can be stronger than LED lighting. V2: This section needs to be made clearer, for example why is light halation caused by LEDs, but not by EL wire? Is there a reference to support this?
Response 8: There is no literature that investigates the appearance of light between LEDs and EL wires as far as the we know. In addition, if the LED light is photographed with a camera, the way it looks will change due to the influence of the equipment, hence it is not possible to provide a video or photo as a reference. Therefore, it is as a future work.
When using LEDs, since the light from each LED is seen directly, the light from each LED diffuses and overlaps. For this reason, the LEDs placed on the fingertips overlap with each other, and the movement of each finger cannot be understood. However, when using EL wires, we are looking indirectly at the light passing through the wire, so the light is difficult to diffuse and can be seen as a wire shape. Therefore, you can see the movement from the LED even if you put it on your fingertip.
193 A reference to back up this statement would be good. What is the limitation? V2: No reference added.
Response 9: Although there are no actual tracking cases, for example, kinect can only track up to 6 people. Therefore, to track 100 dancers more than 17 kinects and a large tracking system are required, which is not prctical.
we added a reference in line 215.
203 are not mare. V2: Should read ‘most are one-off events’
Response 10: we changed to this word.
249 What is meant by a 'proper position'? V2: An explanation has not been provided.
Response 11: we added explanation in line 273-274.
266 Best to add some detail of the different demands that a dance performance places on a costume. V2: Best to explain what happens to the costumes that is different between an entertainment environment and other environments.
Response 12: In the entertainment environment, the production effect is given the highest priority, therefore the Proxemics and human movement that are prioritized in other environments are often sacrificed, which enforces efforts to dancers. therefore, design guidelines for entertainment environment are needed.
We added in line 294-295.
273 Repetition of the need to keep secrets. Best to leave this information out and present the work that can be published without causing IP issues. V2: There is still unnecessary repetition of the need to maintain IP. The basic details of the LED units and associated circuitry must be provided. The reader needs to know what technology was being used in order for the paper to be useful to anyone except the authors. There are very many different types of LEDs on the market, and they are placed together in lighting strips, lamps and many other types of assembly. Different LED units will respond differently to the stresses and strains of performance, so the paper needs to state which generic type of LED unit was used; what type of material the LEDs were placed in to hold the unit together; and how wiring was used to connect the units. For example are the LEDs held in a rubber or thermoset plastic substrate? Details such as these will make a great difference to the way in which the LED units break. Some detail is now supplied below Fig 2. Including this in a more comprehensive ‘materials’ or similar section at the start of the paper would make it easier for the reader to understand what was being tested. A scientific paper must contain information that will enable the reader to repeat the experiments that are described. Without this information, there is insufficient scientific rigour for the paper to be published. (It is not necessary to publish details of the performances.)
Response 13: we added the explanation of LED units in line 333-338.
293 Give some overall description of the costume before going into details of separate parts. Describe what changes were made for each update, and how this affected the LED failure rate.
V2: Some detail of the costume is needed: for example was it always a body suit with LED units sewn or glued on? If so, what type of body suit made from what material? This information should be included in with the details of the LED units. A very basic diagram of the costume, showing what is included where, including battery placement, would enable the reader to understand what exactly is being discussed, and could be created without infringing IP.
Response 14: we added device and battery positions and explanation about LED units and how to store LED units in the costume.
Figure 2: What is meant by ‘ring-shaped’ LED? Is this several individual LEDs in a ring structure? Precise details of this part are required.
V2: no details supplied. The reader needs to know the basics: how many LEDs of what size are held together and connected how?
Response 16: ring-shaped addressable LED strip. It consists of 16 LEDs and is arranged in a ring shape. LED source is SMD5050 built-in SK6812, and outer diameter is 45mm, inner diameter is 30mm, and plug housing JST SM connector is soldered to input and receptacle housing JST SM connector is soldered to output. PCB thickness is 1mm.
We added the explanation.
300 Give full details of the LED strips, including manufacturer. V2: See my comments for line 293. The reader needs to know the basic details of the type of technology being used, otherwise the paper is not scientifically sound. An ‘SM connector’ is now described in the new version of the paper. This needs a fuller explanation: what is an SM connector and where is it on the costume? How does it connect one unit to the next or to the battery? This information could be included without disclosing precise details, but a description of ‘LED units’ that are made of unknown materials and are held on a costume in unknown ways are insufficiently rigorous to be published.
Response 17: We added the explanation in line 333-338.
307 Give precise details of the lengths of the LED strips: longer and shorter ones. V2 47 cm length now supplied. What were the dimensions of each LED unit used in the costume? How does 47 cm relate to the other units. Please add this to the generic details of the costume and LED units that need to be included in a ‘materials’ section.
Response 18: The average length of the LED unit used in the costume is approximately 16 cm and the second longest LED unit is approximately 38 cm. therefore, too long LED is easy to break.
We added the explanation in line 352-353.
336 Do you mean LEDs or LED units that broke? V2: Question not answered.
Response 19: Yes I do.
351 How do you know that the heat and sweat of the dancer caused this type of breakage? V2: Question not answered.
Response 20: These LEDs had water drops between the parts. Once LED is submerged, it is easy to peel off from the circuit board and deteriorates quickly.
352 What is meant by rending? More detail is needed about the way in which the cable got caught? V2: No explanation supplied.
Response 21:In this paper, "rending" means "tear-off". Each LED unit is connected with a cable, but if the cable is entangled or does not fit in the specified location, it will be tear-off during the dance.
we changed "rending" to "tear-off" and added the explanation in line 401.
Figure 3. Make the caption clearer: This is the front, rear, side of the LED strip, not the positions on the dancer as in Figure 4. V2: Amendment not made to Fig. 3 caption.
Response 22: We changed the caption of figure 3.
360 The costume is symmetrical, but dance rarely is! Better to say that only one side of the costume was assessed due to ?time constraints. This is a limitation to the study. V2: no explanation supplied.
Response 23: we added the explantion in line 409-410.
371 Was the endurance test based on any previous work, or on standards or guidelines? V2: no explanation supplied.
Response 24: We added the explantion in line 424-425.
Figure 5: More detail in the caption will help the reader to understand how the diagrams on the left relate to the photos on the right. V2: No amendment supplied.
Response 25: We added the detail in the caption.
392 Why were different amounts of repetitions of the test chosen for different body parts? V2: No clarification supplied.
Response 26: we added the explantion in line 447-448.
Tables 2 – 4: It would be easier to understand how these related to the photos in Figure 5 if there were some more photos or diagrams alongside the tables. V2: no diagrams have been supplied.
Response 27: We added diagrams next to the tables.
406 Explain how the finding about LED unit fracture relates to the points that are in the tables (Af etc). V2: No changes have been made.
Response 28: In this test, we found that the LED unit located at the side of the joints(Ae, Ce, Ak, Bk, Fk, and Gk) and around the groin(from Ak to Ff) easily.
477 How is the ‘more complex’ wiring constructed and arranged on the costume? V2: Please make the change in the number of LED units clearer. How many LED units were on the previous version of the costume. Details of the costume are supplied in this section. They should also be in the materials and methods section, so that the reader knows what type of costume is being described early in the paper.
Response 30: we added the number of LED units on the previous costumes and the detail of new LED costume.
489 How does the polycarbonate affect the dancers? V2: Insufficient detail given. ‘we investigated various reinforcement materials’ is too vague for a scientific paper. Which materials? How was the flexibility of polycarbonate assessed? The type of polycarbonate used needs to be stated, as the rigidity of polycarbonate is variable: https://www.acplasticsinc.com/informationcenter/r/acrylic-vs-polycarbonate
Response 32: We investigated 3 acrylic plate that thickness are 2 mm, 3 mm, and 5 mm. 2 mm and 3 mm thickness plates is too soft to reinforce LED strip. Moreover, the 5 mm thickness acrylic plate cracks when bent too much. If it breaks, it is possible that the debris hurt the dancer, therefore it is dangerous to use it for costumes. On the other hand, we investigated the GP polycarbonate plates that thickness are 2 mm, 3 mm, and 5 mm. 3 mm and 5 mm thickness plates are too hard to bend them, which disturb dancing. 2 mm thickness plate bends moderately and does not interfere with dance. Therefore, polycarbonate plate that thickness is 2 mm is selected.